# Scaled Least Squares Estimator for GLMs in Large-Scale Problems

**Murat A. Erdogdu**
Department of Statistics
Stanford University
erdogdu@stanford.edu

**Mohsen Bayati**
Graduate School of Business
Stanford University
bayati@stanford.edu

**Lee H. Dicker**
Department of Statistics and Biostatistics
Rutgers University and Amazon *
ldicker@stat.rutgers.edu

## Abstract

We study the problem of efficiently estimating the coefficients of generalized linear models (GLMs) in the large-scale setting where the number of observations $n$ is much larger than the number of predictors $p$, i.e. $n \gg p \gg 1$. We show that in GLMs with random (not necessarily Gaussian) design, the GLM coefficients are approximately proportional to the corresponding ordinary least squares (OLS) coefficients. Using this relation, we design an algorithm that achieves the same accuracy as the maximum likelihood estimator (MLE) through iterations that attain up to a cubic convergence rate, and that are cheaper than any batch optimization algorithm by at least a factor of $\mathcal{O}(p)$. We provide theoretical guarantees for our algorithm, and analyze the convergence behavior in terms of data dimensions. Finally, we demonstrate the performance of our algorithm through extensive numerical studies on large-scale real and synthetic datasets, and show that it achieves the highest performance compared to several other widely used optimization algorithms.

## 1 Introduction

We consider the problem of efficiently estimating the coefficients of generalized linear models (GLMs) when the number of observations $n$ is much larger than the dimension of the coefficient vector $p$, ($n \gg p \gg 1$). GLMs play a crucial role in numerous machine learning and statistics problems, and provide a miscellaneous framework for many regression and classification tasks. Celebrated examples include ordinary least squares, logistic regression, multinomial regression and many applications involving graphical models [MN89, WJ08, KF09].

The standard approach to estimating the regression coefficients in a GLM is the maximum likelihood method. Under standard assumptions on the link function, the maximum likelihood estimator (MLE) can be written as the solution to a convex minimization problem [MN89]. Due to the non-linear structure of the MLE problem, the resulting optimization task requires iterative methods. The most commonly used optimization technique for computing the MLE is the Newton-Raphson method, which may be viewed as a reweighted least squares algorithm [MN89]. This method uses a second order approximation to benefit from the curvature of the log-likelihood and achieves locally quadratic convergence. A drawback of this approach is its excessive per-iteration cost of $\mathcal{O}(np^2)$. To remedy this, Hessian-free Krylov sub-space based methods such as conjugate gradient and minimal residual are used, but the resulting direction is imprecise [HS52, PS75, Mar10]. On the other hand, first order

approximation yields the gradient descent algorithm, which attains a linear convergence rate with $\mathcal{O}(np)$ per-iteration cost. Although its convergence rate is slow compared to that of second order methods, its modest per-iteration cost makes it practical for large-scale problems. In the regime $n \gg p$, another popular optimization technique is the class of Quasi-Newton methods [Bis95, Nes04], which can attain a per-iteration cost of $\mathcal{O}(np)$, and the convergence rate is locally super-linear; a well-known member of this class of methods is the BFGS algorithm [Nes04]. There are recent studies that exploit the special structure of GLMs [Erd15], and achieve near-quadratic convergence with a per-iteration cost of $\mathcal{O}(np)$, and an additional cost of covariance estimation.

In this paper, we take an alternative approach to fitting GLMs, based on an identity that is well-known in some areas of statistics, but appears to have received relatively little attention for its computational implications in large scale problems. Let $\beta^{\mathsf{glm}}$ denote the GLM regression coefficients, and let $\beta^{\mathsf{ols}}$ denote the corresponding ordinary least squares (OLS) coefficients (this notation will be defined more precisely in Section 2). Then, under certain random predictor (design) models,

$$\beta^{\mathsf{glm}} \propto \beta^{\mathsf{ols}}. \tag{1}$$

For logistic regression with Gaussian design (which is equivalent to Fisher's discriminant analysis), (1) was noted by Fisher in the 1930s [Fis36]; a more general formulation for models with Gaussian design is given in [Bri82]. The relationship (1) suggests that if the constant of proportionality is known, then $\beta^{\mathsf{glm}}$ can be estimated by computing the OLS estimator, which may be substantially simpler than finding the MLE for the original GLM. Our work in this paper builds on this idea.

Our contributions can be summarized as follows.

1. We show that $\beta^{\mathsf{glm}}$ is approximately proportional to $\beta^{\mathsf{ols}}$ in random design GLMs, regardless of the predictor distribution. That is, we prove

$$\left\| \beta^{\mathsf{glm}} - c_\Psi \times \beta^{\mathsf{ols}} \right\|_\infty \lesssim \frac{1}{p}, \text{ for some } c_\Psi \in \mathbb{R}.$$

2. We design a computationally efficient estimator for $\beta^{\mathsf{glm}}$ by first estimating the OLS coefficients, and then estimating the proportionality constant $c_\Psi$. We refer to the resulting estimator as the Scaled Least Squares (SLS) estimator and denote it by $\hat{\beta}^{\mathsf{sls}}$. After estimating the OLS coefficients, the second step of our algorithm involves finding a root of a real valued function; this can be accomplished using iterative methods with up to a cubic convergence rate and only $\mathcal{O}(n)$ per-iteration cost. This is cheaper than the classical batch methods mentioned above by at least a factor of $\mathcal{O}(p)$.

3. For random design GLMs with sub-Gaussian predictors, we show that

$$\left\| \hat{\beta}^{\mathsf{sls}} - \beta^{\mathsf{glm}} \right\|_\infty \lesssim \frac{1}{p} + \sqrt{\frac{p}{n / \max\{\log(n), p\}}}.$$

This bound characterizes the performance of the proposed estimator in terms of data dimensions, and justifies the use of the algorithm in the regime $n \gg p \gg 1$.

4. We study the statistical and computational performance of $\hat{\beta}^{\mathsf{sls}}$, and compare it to that of the MLE (using several well-known implementations), on a variety of large-scale datasets.

The rest of the paper is organized as follows: Section 1.1 surveys the related work and Section 2 introduces the required background and the notation. In Section 3, we provide the intuition behind the relationship (1), which are based on exact calculations for GLMs with Gaussian design. In Section 4, we propose our algorithm and discuss its computational properties. Section 5 provides a thorough comparison between the proposed algorithm and other existing methods. Theoretical results may be found in Section 6. Finally, we conclude with a brief discussion in Section 7.

## 1.1 Related work

As mentioned in Section 1, the relationship (1) is well-known in several forms in statistics. Brillinger [Bri82] derived (1) for models with Gaussian predictors. Li & Duan [LD89] studied model misspecification problems in statistics and derived (1) when the predictor distribution has linear conditional means (this is a slight generalization of Gaussian predictors). More recently, Stein's lemma [BEM13] and the relationship (1) has been revisited in the context of compressed sensing [PV15, TAH15], where it has been shown that the standard lasso estimator may be very effective when used in models

where the relationship between the expected response and the signal is nonlinear, and the predictors (i.e. the design or sensing matrix) are Gaussian. A common theme for all of this previous work is that it focuses solely on settings where (1) holds exactly and the predictors are Gaussian (or, in [LD89], nearly Gaussian). Two key novelties of the present paper are (i) our focus on the computational benefits following from (1) for large scale problems with $n \gg p \gg 1$; and (ii) our rigorous analysis of models with non-Gaussian predictors, where (1) is shown to be approximately valid.

## 2   Preliminaries and notation

We assume a random design setting, where the observed data consists of $n$ random iid pairs $(y_1, x_1)$, $(y_2, x_2)$, ..., $(y_n, x_n)$; $y_i \in \mathbb{R}$ is the response variable and $x_i = (x_{i1}, \ldots, x_{ip})^T \in \mathbb{R}^p$ is the vector of predictors or covariates. We focus on problems where fitting a GLM is desirable, but we do not need to assume that $(y_i, x_i)$ are actually drawn from the corresponding statistical model (i.e. we allow for model misspecification).

The MLE for GLMs with canonical link is defined by

$$\hat{\beta}^{\mathsf{mle}} = \underset{\beta \in \mathbb{R}^p}{\operatorname{argmax}} \ \frac{1}{n} \sum_{i=1}^n y_i \langle x_i, \beta \rangle - \Psi(\langle x_i, \beta \rangle). \tag{2}$$

where $\langle \cdot, \cdot \rangle$ denotes the Euclidean inner-product on $\mathbb{R}^p$, and $\Psi$ is a sufficiently smooth convex function. The GLM coefficients $\beta^{\mathsf{glm}}$ are defined by taking the population average in (2):

$$\beta^{\mathsf{glm}} = \underset{\beta \in \mathbb{R}^p}{\operatorname{argmax}} \ \mathbb{E}\left[y_i \langle x_i, \beta \rangle - \Psi(\langle x_i, \beta \rangle)\right]. \tag{3}$$

While we make no assumptions on $\Psi$ beyond smoothness, note that if $\Psi$ is the cumulant generating function for $y_i \mid x_i$, then we recover the standard GLM with canonical link and regression parameters $\beta^{\mathsf{glm}}$ [MN89]. Examples of GLMs in this form include logistic regression, with $\Psi(w) = \log\{1 + e^w\}$; Poisson regression, with $\Psi(w) = e^w$; and linear regression (least squares), with $\Psi(w) = w^2/2$.

Our objective is to find a computationally efficient estimator for $\beta^{\mathsf{glm}}$. The alternative estimator for $\beta^{\mathsf{glm}}$ proposed in this paper is related to the OLS coefficient vector, which is defined by $\beta^{\mathsf{ols}} := \mathbb{E}[x_i x_i^T]^{-1} \mathbb{E}[x_i y_i]$; the corresponding OLS estimator is $\hat{\beta}^{\mathsf{ols}} := (\mathbf{X}^T \mathbf{X})^{-1} \mathbf{X}^T y$, where $\mathbf{X} = (x_1, \ldots, x_n)^T$ is the $n \times p$ design matrix and $y = (y_1, \ldots, y_n)^T \in \mathbb{R}^n$.

Additionally, throughout the text we let $[m] = \{1, 2, ..., m\}$, for positive integers $m$, and we denote the size of a set $S$ by $|S|$. The $m$-th derivative of a function $g : \mathbb{R} \to \mathbb{R}$ is denoted by $g^{(m)}$. For a vector $u \in \mathbb{R}^p$ and a $n \times p$ matrix $\mathbf{U}$, we let $\|u\|_q$ and $\|\mathbf{U}\|_q$ denote the $\ell_q$-vector and -operator norms, respectively. If $S \subseteq [n]$, let $\mathbf{U}_S$ denote the $|S| \times p$ matrix obtained from $\mathbf{U}$ by extracting the rows that are indexed by $S$. For a symmetric matrix $\mathbf{M} \in \mathbb{R}^{p \times p}$, $\lambda_{\max}(\mathbf{M})$ and $\lambda_{\min}(\mathbf{M})$ denote the maximum and minimum eigenvalues, respectively. $\rho_k(\mathbf{M})$ denotes the condition number of $\mathbf{M}$ with respect to $k$-norm. We denote by $\mathsf{N}_q$ the $q$-variate normal distribution.

## 3   OLS is equivalent to GLM up to a scalar factor

To motivate our methodology, we assume in this section that the covariates are multivariate normal, as in [Bri82]. These distributional assumptions will be relaxed in Section 6.

**Proposition 1.** *Assume that the covariates are multivariate normal with mean 0 and covariance matrix $\Sigma = \mathbb{E}\left[x_i x_i^T\right]$, i.e. $x_i \sim \mathsf{N}_p(0, \Sigma)$. Then $\beta^{\mathsf{glm}}$ can be written as*

$$\beta^{\mathsf{glm}} = c_\Psi \times \beta^{\mathsf{ols}},$$

*where $c_\Psi \in \mathbb{R}$ satisfies the equation $1 = c_\Psi \mathbb{E}\left[\Psi^{(2)}(\langle x, \beta^{\mathsf{ols}}\rangle c_\Psi)\right]$.*

*Proof of Proposition 1.* The optimal point in the optimization problem (3), has to satisfy the following normal equations,

$$\mathbb{E}\left[y_i x_i\right] = \mathbb{E}\left[x_i \Psi^{(1)}(\langle x_i, \beta \rangle)\right]. \tag{4}$$

Now, denote by $\phi(x \mid \Sigma)$ the multivariate normal density with mean 0 and covariance matrix $\Sigma$. We recall the well-known property of Gaussian density $\mathrm{d}\phi(x \mid \Sigma)/\mathrm{d}x = -\Sigma^{-1} x \phi(x \mid \Sigma)$. Using this

---

**Algorithm 1** SLS: Scaled Least Squares Estimator

---

**Input:** Data $(y_i, x_i)_{i=1}^n$

**Step 1. Compute the least squares estimator:** $\hat{\beta}^{\text{ols}}$ **and** $\hat{y} = \mathbf{X}\hat{\beta}^{\text{ols}}$.
   For a sub-sampling based OLS estimator, let $S \subset [n]$ be a
   random subset and take $\hat{\beta}^{\text{ols}} = \frac{|S|}{n}(\mathbf{X}_S^T \mathbf{X}_S)^{-1}\mathbf{X}^T y$.

**Step 2. Solve the following equation for $c \in \mathbb{R}$:** $1 = \frac{c}{n}\sum_{i=1}^n \Psi^{(2)}(c\,\hat{y}_i)$.
   Use Newton's root-finding method:
      Initialize $c = 2/\text{Var}(y_i)$;
      Repeat until convergence:

$$c \leftarrow c - \frac{c\frac{1}{n}\sum_{i=1}^n \Psi^{(2)}(c\,\hat{y}_i) - 1}{\frac{1}{n}\sum_{i=1}^n \left\{\Psi^{(2)}(c\,\hat{y}_i) + c\,\Psi^{(3)}(c\,\hat{y}_i)\right\}}.$$

**Output**: $\hat{\beta}^{\text{sls}} = c \times \hat{\beta}^{\text{ols}}$.

---

and integration by parts on the right hand side of the above equation, we obtain

$$\mathbb{E}\left[x_i \Psi^{(1)}(\langle x_i, \beta \rangle)\right] = \int x \Psi^{(1)}(\langle x, \beta \rangle)\phi(x \mid \mathbf{\Sigma})\,\mathrm{d}x = \mathbf{\Sigma}\beta\mathbb{E}\left[\Psi^{(2)}(\langle x_i, \beta \rangle)\right] \qquad (5)$$

(this is basically the Stein's lemma). Combining this with the identity (4), we conclude the proof. $\quad\square$

Proposition 1 and its proof provide the main intuition behind our proposed method. Observe that in our derivation, we only worked with the right hand side of the normal equations (4) which does not depend on the response variable $y_i$. The equivalence holds regardless of the joint distribution of $(y_i, x_i)$, whereas in [Bri82], $y_i$ is assumed to follow a single index model. In Section 6, where we extend the method to non-Gaussian predictors, (5) is generalized via the zero-bias transformations.

### 3.1 Regularization

A version of Proposition 1 incorporating regularization — an important tool for datasets where $p$ is large relative to $n$ or the predictors are highly collinear — is also possible, as outlined briefly in this section. We focus on $\ell^2$-regularization (ridge regression) in this section; some connections with lasso ($\ell^1$-regularization) are discussed in Section 6 and Corollary 1.

For $\lambda \geq 0$, define the $\ell_2$-regularized GLM coefficients,

$$\beta_\lambda^{\text{glm}} = \underset{\beta \in \mathbb{R}^p}{\text{argmax}}\ \mathbb{E}\left[y_i\langle x_i, \beta \rangle - \Psi(\langle x_i, \beta \rangle)\right] - \frac{\lambda}{2}\|\beta\|_2^2 \qquad (6)$$

and the corresponding $\ell^2$-regularized OLS coefficients $\beta_\lambda^{\text{ols}} = \left(\mathbb{E}\left[x_i x_i^T\right] + \lambda \mathbf{I}\right)^{-1}\mathbb{E}\left[x_i y_i\right]$ (so $\beta^{\text{glm}} = \beta_0^{\text{glm}}$ and $\beta^{\text{ols}} = \beta_0^{\text{ols}}$). The same argument as above implies that

$$\beta_\lambda^{\text{glm}} = c_\Psi \times \beta_\gamma^{\text{ols}}, \quad \text{where} \ \ \gamma = \lambda c_\Psi. \qquad (7)$$

This suggests that the ordinary ridge regression for the linear model can be used to estimate the $\ell^2$-regularized GLM coefficients $\beta_\lambda^{\text{glm}}$. Further pursuing these ideas for problems where regularization is a critical issue may be an interesting area for future research.

## 4   SLS: Scaled Least Squares estimator for GLMs

Motivated by the results in the previous section, we design a computationally efficient algorithm for any GLM task that is as simple as solving the least squares problem; it is described in Algorithm 1. The algorithm has two basic steps. First, we estimate the OLS coefficients, and then in the second step we estimate the proportionality constant via a simple root-finding algorithm.

There are numerous fast optimization methods to solve the least squares problem, and even a superficial review of these could go beyond the page limits of this paper. We emphasize that this step (finding the OLS estimator) does not have to be iterative and it is the main computational cost of the proposed algorithm. We suggest using a sub-sampling based estimator for $\hat{\beta}^{\text{ols}}$, where we only use a subset of the observations to estimate the covariance matrix. Let $S \subset [n]$ be a

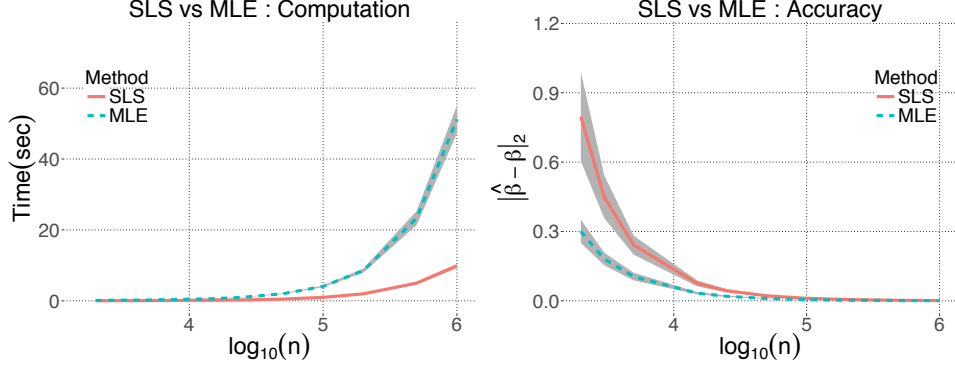

Figure 1: Logistic regression with general Gaussian design. The left plot shows the computational cost (time) for finding the MLE and SLS as $n$ grows and $p = 200$. The right plot depicts the accuracy of the estimators. In the regime where the MLE is expensive to compute, the SLS is found much more rapidly and has the same accuracy. R's built-in functions are used to find the MLE.

random sub-sample and denote by $\mathbf{X}_S$ the sub-matrix formed by the rows of $\mathbf{X}$ in $S$. Then the sub-sampled OLS estimator is given as $\hat{\beta}^{\text{ols}} = \left(\frac{1}{|S|}\mathbf{X}_S^T\mathbf{X}_S\right)^{-1}\frac{1}{n}\mathbf{X}^T y$. Properties of this estimator have been well-studied [Ver10, DLFU13, EM15]. For sub-Gaussian covariates, it suffices to use a sub-sample size of $\mathcal{O}\left(p\log(p)\right)$ [Ver10]. Hence, this step requires a single time computational cost of $\mathcal{O}\left(|S|p^2 + p^3 + np\right) \approx \mathcal{O}\left(p\max\{p^2\log(p), n\}\right)$. For other approaches, we refer reader to [RT08, DLFU13] and the references therein.

The second step of Algorithm 1 involves solving a simple root-finding problem. As with the first step of the algorithm, there are numerous methods available for completing this task. Newton's root-finding method with quadratic convergence or Halley's method with cubic convergence may be appropriate choices. We highlight that this step costs only $\mathcal{O}\left(n\right)$ per-iteration and that we can attain up to a cubic rate of convergence. The resulting per-iteration cost is cheaper than other commonly used batch algorithms by at least a factor of $\mathcal{O}\left(p\right)$ — indeed, the cost of computing the gradient is $\mathcal{O}\left(np\right)$. For simplicity, we use Newton's root-finding method initialized at $c = 2/\text{Var}\left(y_i\right)$. Assuming that the GLM is a good approximation to the true conditional distribution, by the law of total variance and basic properties of GLMs, we have

$$\text{Var}\left(y_i\right) = \mathbb{E}\left[\text{Var}\left(y_i \mid x_i\right)\right] + \text{Var}\left(\mathbb{E}\left[y_i \mid x_i\right]\right) \approx c_\Psi^{-1} + \text{Var}\left(\Psi^{(1)}(\langle x_i, \beta\rangle)\right). \tag{8}$$

It follows that this initialization is reasonable as long as $c_\Psi^{-1} \approx \mathbb{E}\left[\text{Var}\left(y_i \mid x_i\right)\right]$ is not much smaller than $\text{Var}\left(\Psi^{(1)}(\langle x_i, \beta\rangle)\right)$. Our experiments show that SLS is very robust to initialization.

In Figure 1, we compare the performance of our SLS estimator to that of the MLE, when both are used to analyze synthetic data generated from a logistic regression model under general Gaussian design with randomly generated covariance matrix. The left plot shows the computational cost of obtaining both estimators as $n$ increases for fixed $p$. The right plot shows the accuracy of the estimators. In the regime $n \gg p \gg 1$ — where the MLE is hard to compute — the MLE and the SLS achieve the same accuracy, yet SLS has significantly smaller computation time. We refer the reader to Section 6 for theoretical results characterizing the finite sample behavior of the SLS.

## 5 Experiments

This section contains the results of a variety of numerical studies, which show that the Scaled Least Squares estimator reaches the minimum achievable test error substantially faster than commonly used batch algorithms for finding the MLE. Both logistic and Poisson regression models (two types of GLMs) are utilized in our analyses, which are based on several synthetic and real datasets.

Below, we briefly describe the optimization algorithms for the MLE that were used in the experiments.

1. **Newton-Raphson (NR)** achieves locally quadratic convergence by scaling the gradient by the inverse of the Hessian evaluated at the current iterate. Computing the Hessian has a per-iteration cost of $\mathcal{O}\left(np^2\right)$, which makes it impractical for large-scale datasets.
2. **Newton-Stein (NS)** is a recently proposed second-order batch algorithm specifically designed for GLMs [Erd16]. The algorithm uses Stein's lemma and sub-sampling to efficiently estimate the Hessian with $\mathcal{O}\left(np\right)$ per-iteration cost, achieving near quadratic rates.

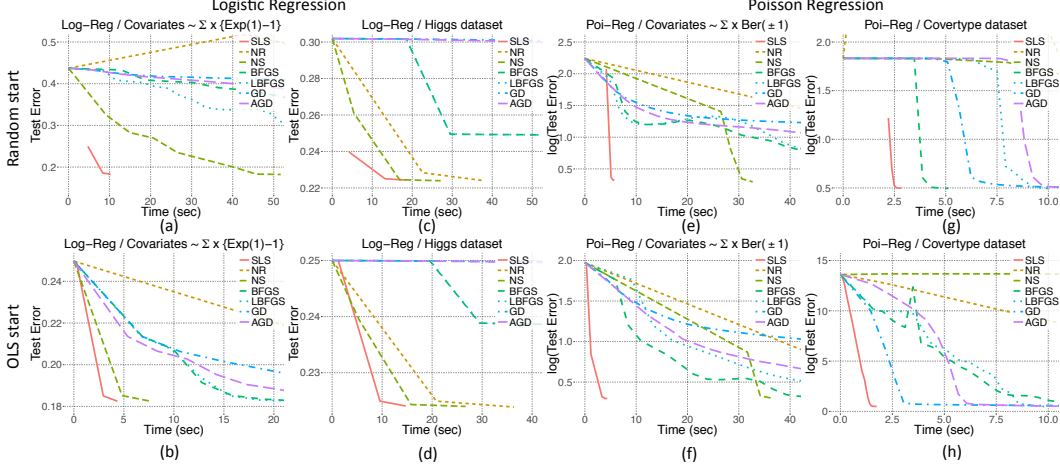

Figure 2: Performance of SLS compared to that of MLE obtained with various optimization algorithms on several datasets. SLS is represented with red straight line. The details are provided in Table 1.

3. **Broyden-Fletcher-Goldfarb-Shanno (BFGS)** is the most popular and stable quasi-Newton method [Nes04]. At each iteration, the gradient is scaled by a matrix that is formed by accumulating information from previous iterations and gradient computations. The convergence is locally super-linear with a per-iteration cost of $\mathcal{O}(np)$.

4. **Limited memory BFGS (LBFGS)** is a variant of BFGS, which uses only the recent iterates and gradients to approximate the Hessian, providing significant improvement in terms of memory usage. LBFGS has many variants; we use the formulation given in [Bis95].

5. **Gradient descent (GD)** takes a step in the opposite direction of the gradient, evaluated at the current iterate. Its performance strongly depends on the condition number of the design matrix. Under certain assumptions, the convergence is linear with $\mathcal{O}(np)$ per-iteration cost.

6. **Accelerated gradient descent (AGD)** is a modified version of gradient descent with an additional "momentum" term [Nes83]. Its per iteration cost is $\mathcal{O}(np)$ and its performance strongly depends on the smoothness of the objective function.

For all the algorithms, the step size at each iteration is chosen via the backtracking line search [BV04].

Recall that the proposed Algorithm 1 is composed of two steps; the first finds an estimate of the OLS coefficients. This up-front computation is not needed for any of the MLE algorithms described above. On the other hand, each of the MLE algorithms requires some initial value for $\beta$, but no such initialization is needed to find the OLS estimator in Algorithm 1. This raises the question of how the MLE algorithms should be initialized, in order to compare them fairly with the proposed method. We consider two scenarios in our experiments: first, we use the OLS estimator computed for Algorithm 1 to initialize the MLE algorithms; second, we use a random initial value.

On each dataset, the main criterion for assessing the performance of the estimators is how rapidly the minimum test error is achieved. The test error is measured as the mean squared error of the estimated mean using the current parameters at each iteration on a test dataset, which is a randomly selected (and set-aside) 10% portion of the entire dataset. As noted previously, the MLE is more accurate for small $n$ (see Figure 1). However, in the regime considered here ($n \gg p \gg 1$), the MLE and the SLS perform very similarly in terms of their error rates; for instance, on the Higgs dataset, the SLS and MLE have test error rates of $22.40\%$ and $22.38\%$, respectively. For each dataset, the minimum achievable test error is set to be the maximum of the final test errors, where the maximum is taken over all of the estimation methods. Let $\boldsymbol{\Sigma}^{(1)}$ and $\boldsymbol{\Sigma}^{(2)}$ be two randomly generated covariance matrices. The datasets we analyzed were: (i) a synthetic dataset generated from a logistic regression model with iid $\{\text{exponential}(1)-1\}$ predictors scaled by $\boldsymbol{\Sigma}^{(1)}$; (ii) the Higgs dataset (logistic regression) [BSW14]; (iii) a synthetic dataset generated from a Poisson regression model with iid binary($\pm 1$) predictors scaled by $\boldsymbol{\Sigma}^{(2)}$; (iv) the Covertype dataset (Poisson regression) [BD99].

In all cases, the SLS outperformed the alternative algorithms for finding the MLE by a large margin, in terms of computation. Detailed results may be found in Figure 2 and Table 1. We provide additional experiments with different datasets in the Supplementary Material.

Table 1: Details of the experiments shown in Figure 2.

| MODEL | LOGISTIC REGRESSION | | | | POISSON REGRESSION | | | |
|---|---|---|---|---|---|---|---|---|
| DATASET | $\Sigma\times\{\text{Exp(1)-1}\}$ | | Higgs [BSW14] | | $\Sigma\times\text{Ber}(\pm1)$ | | Covertype [BD99] | |
| SIZE | $n=6.0\times10^5, p=300$ | | $n=1.1\times10^7, p=29$ | | $n=6.0\times10^5, p=300$ | | $n=5.8\times10^5, p=53$ | |
| INITIALIZED | Rnd | Ols | Rnd | Ols | Rnd | Ols | Rnd | Ols |
| PLOT | (A) | (B) | (C) | (D) | (E) | (F) | (G) | (H) |
| METHOD | TIME IN SECONDS / NUMBER OF ITERATIONS (TO REACH MIN TEST ERROR) | | | | | | | |
| SLS | 8.34/4 | 2.94/3 | 13.18/3 | 9.57/3 | 5.42/5 | 3.96/5 | 2.71/6 | 1.66/20 |
| NR | 301.06/6 | 82.57/3 | 37.77/3 | 36.37/3 | 170.28/5 | 130.1/4 | 16.7/8 | 32.48/18 |
| NS | 51.69/8 | 7.8/3 | 27.11/4 | 26.69/4 | 32.71/5 | 36.82/4 | 21.17/10 | 282.1/216 |
| BFGS | 148.43/31 | 24.79/8 | 660.92/68 | 701.9/68 | 67.24/29 | 72.42/26 | 5.12/7 | 22.74/59 |
| LBFGS | 125.33/39 | 24.61/8 | 6368.1/651 | 6946.1/670 | 224.6/106 | 357.1/88 | 10.01/14 | 10.05/17 |
| GD | 669/138 | 134.91/25 | 100871/10101 | 141736/13808 | 1711/513 | 1364/374 | 14.35/25 | 33.58/87 |
| AGD | 218.1/61 | 35.97/12 | 2405.5/251 | 2879.69/277 | 103.3/51 | 102.74/40 | 11.28/15 | 11.95/25 |

## 6 Theoretical results

In this section, we use the zero-bias transformations [GR97] to generalize the equivalence between OLS and GLMs to settings where the covariates are non-Gaussian.

**Definition 1.** *Let $z$ be a random variable with mean 0 and variance $\sigma^2$. Then, there exists a random variable $z^*$ that satisfies $\mathbb{E}[zf(z)] = \sigma^2\mathbb{E}[f^{(1)}(z^*)]$, for all differentiable functions $f$. The distribution of $z^*$ is said to be the z-zero-bias distribution.*

The existence of $z^*$ in Definition 1 is a consequence of Riesz representation theorem [GR97]. The normal distribution is the unique distribution whose zero-bias transformation is itself (i.e. the normal distribution is a fixed point of the operation mapping the distribution of $z$ to that of $z^*$).

To provide some intuition behind the usefulness of the zero-bias transformation, we refer back to the proof of Proposition 1. For simplicity, assume that the covariate vector $x_i$ has iid entries with mean 0, and variance 1. Then the zero-bias transformation applied to the $j$-th normal equation in (4) yields

$$\underbrace{\mathbb{E}[y_ix_{ij}] = \mathbb{E}\left[x_{ij}\Psi^{(1)}\big(x_{ij}\beta_j + \Sigma_{k\neq j}x_{ik}\beta_k\big)\right]}_{j\text{-th normal equation}} = \underbrace{\beta_j\mathbb{E}\left[\Psi^{(2)}\big(x_{ij}^*\beta_j + \Sigma_{k\neq j}x_{ik}\beta_{ik}\big)\right]}_{\text{Zero-bias transformation}}. \quad (9)$$

The distribution of $x_{ij}^*$ is the $x_{ij}$-zero-bias distribution and is entirely determined by the distribution of $x_{ij}$; general properties of $x_{ij}^*$ can be found, for example, in [CGS10]. If $\beta$ is well spread, it turns out that taken together, with $j = 1, \ldots, p$, the far right-hand side in (9) behaves similar to the right side of (5), with $\Sigma = \mathbf{I}$; that is, the behavior is similar to the Gaussian case, where the proportionality relationship given in Proposition 1 holds. This argument leads to an approximate proportionality relationship for non-Gaussian predictors, which, when carried out rigorously, yields the following.

**Theorem 1.** *Suppose that the covariate vector $x_i$ has mean 0 and covariance matrix $\Sigma$ and, furthermore, that the random vector $\Sigma^{-1/2}x_i$ has independent entries and its sub-Gaussian norm is bounded by $\kappa$. Assume that the function $\Psi^{(2)}$ is Lipschitz continuous with constant $k$. Let $\|\beta\|_2 = \tau$ and assume $\beta$ is $r$-well-spread in the sense that $\tau/\|\beta\|_\infty = r\sqrt{p}$ for some $r \in (0, 1]$. Then, for $c_\Psi = 1/\mathbb{E}\left[\Psi^{(2)}(\langle x_i, \beta^{\text{glm}}\rangle)\right]$, and $\rho = \rho_\infty(\Sigma^{1/2})$ denoting the condition number of $\Sigma^{1/2}$, we have*

$$\left\|\frac{1}{c_\Psi} \times \beta^{\text{glm}} - \beta^{\text{ols}}\right\|_\infty \leq \frac{\eta}{p}, \quad \text{where } \eta = 8k\kappa^3\rho\|\Sigma^{1/2}\|_\infty(\tau/r)^2. \quad (10)$$

Theorem 1 is proved in the Supplementary Material. It implies that the population parameters $\beta^{\text{ols}}$ and $\beta^{\text{glm}}$ are approximately equivalent up to a scaling factor, with an error bound of $\mathcal{O}(1/p)$. The assumption that $\beta^{\text{glm}}$ is well-spread can be relaxed with minor modifications. For example, if we have a sparse coefficient vector, where $\text{supp}(\beta^{\text{glm}}) = \{j; \beta_j^{\text{glm}} \neq 0\}$ is the support set of $\beta^{\text{glm}}$, then Theorem 1 holds with $p$ replaced by the size of the support set.

An interesting consequence of Theorem 1 and the remarks following the theorem is that whenever an entry of $\beta^{\text{glm}}$ is zero, the corresponding entry of $\beta^{\text{ols}}$ has to be small, and conversely. For $\lambda \geq 0$, define the lasso coefficients

$$\beta_\lambda^{\text{lasso}} = \underset{\beta\in\mathbb{R}^p}{\text{argmin}}\ \frac{1}{2}\mathbb{E}\left[(y_i - \langle x_i, \beta\rangle)^2\right] + \lambda\|\beta\|_1. \quad (11)$$

**Corollary 1.** *For any* $\lambda \geq \eta/|\mathrm{supp}(\beta^{\mathsf{glm}})|$, *if* $\mathbb{E}[x_i] = 0$ *and* $\mathbb{E}[x_i x_i^T] = \mathbf{I}$, *we have* $\mathrm{supp}(\beta^{\mathsf{lasso}}) \subset \mathrm{supp}(\beta^{\mathsf{glm}})$. *Further, if* $\lambda$ *and* $\beta^{\mathsf{glm}}$ *also satisfy that* $\forall j \in \mathrm{supp}(\beta^{\mathsf{glm}})$, $|\beta_j^{\mathsf{glm}}| > c_\Psi \left(\lambda + \eta/|\mathrm{supp}(\beta^{\mathsf{glm}})|\right)$, *then we have* $\mathrm{supp}(\beta^{\mathsf{lasso}}) = \mathrm{supp}(\beta^{\mathsf{glm}})$.

So far in this section, we have only discussed properties of the population parameters, such as $\beta^{\mathsf{glm}}$. In the remainder of this section, we turn our attention to results for the estimators that are the main focus of this paper; these results ultimately build on our earlier results, i.e. Theorem 1.

In order to precisely describe the performance of $\hat{\beta}^{\mathsf{sls}}$, we first need bounds on the OLS estimator. The OLS estimator has been studied extensively in the literature; however, for our purposes, we find it convenient to derive a new bound on its accuracy. While we have not seen this exact bound elsewhere, it is very similar to Theorem 5 of [DLFU13].

**Proposition 2.** *Assume that* $\mathbb{E}[x_i] = 0$, $\mathbb{E}[x_i x_i^T] = \mathbf{\Sigma}$, *and that* $\mathbf{\Sigma}^{-1/2} x_i$ *and* $y_i$ *are sub-Gaussian with norms* $\kappa$ *and* $\gamma$, *respectively. For* $\lambda_{min}$ *denoting the smallest eigenvalue of* $\mathbf{\Sigma}$, *and* $|S| > \eta p$,

$$\left\|\hat{\beta}^{\mathsf{ols}} - \beta^{\mathsf{ols}}\right\|_2 \leq \eta \lambda_{min}^{-1/2} \sqrt{\frac{p}{|S|}}, \tag{12}$$

*with probability at least* $1 - 3e^{-p}$, *where* $\eta$ *depends only on* $\gamma$ *and* $\kappa$.

Proposition 2 is proved in the Supplementary Material. Our main result on the performance of $\hat{\beta}^{\mathsf{sls}}$ is given next.

**Theorem 2.** *Let the assumptions of Theorem 1 and Proposition 2 hold with* $\mathbb{E}[\|\mathbf{\Sigma}^{-1/2} x\|_2] = \tilde{\mu}\sqrt{p}$. *Further assume that the function* $f(z) = z\mathbb{E}\left[\Psi^{(2)}(\langle x, \beta^{\mathsf{ols}}\rangle z)\right]$ *satisfies* $f(\bar{c}) > 1 + \bar{\delta}\sqrt{p}$ *for some* $\bar{c}$ *and* $\bar{\delta}$ *such that the derivative of* $f$ *in the interval* $[0, \bar{c}]$ *does not change sign, i.e., its absolute value is lower bounded by* $\upsilon > 0$. *Then, for* $n$ *and* $|S|$ *sufficiently large, we have*

$$\left\|\hat{\beta}^{\mathsf{sls}} - \beta^{\mathsf{glm}}\right\|_\infty \leq \eta_1 \frac{1}{p} + \eta_2 \sqrt{\frac{p}{\min\{n/\log(n), |S|/p\}}}, \tag{13}$$

*with probability at least* $1 - 5e^{-p}$, *where the constants* $\eta_1$ *and* $\eta_2$ *are defined by*

$$\eta_1 = \eta k \bar{c} \kappa^3 \rho \|\mathbf{\Sigma}^{1/2}\|_\infty (\tau/r)^2 \tag{14}$$

$$\eta_2 = \eta \bar{c} \lambda_{min}^{-1/2} \left(1 + \upsilon^{-1} \lambda_{min}^{1/2} \|\beta^{\mathsf{ols}}\|_\infty \max\{(b + k/\tilde{\mu}), k\bar{c}\}\right), \tag{15}$$

*and* $\eta > 0$ *is a constant depending on* $\kappa$ *and* $\gamma$.

Note that the convergence rate of the upper bound in (13) depends on the sum of the two terms, both of which are functions of the data dimensions $n$ and $p$. The first term on the right in (13) comes from Theorem 1, which bounds the discrepancy between $c_\Psi \times \beta^{\mathsf{ols}}$ and $\beta^{\mathsf{glm}}$. This term is small when $p$ is large, and it does not depend on the number of observations $n$.

The second term in the upper bound (13) comes from estimating $\beta^{\mathsf{ols}}$ and $c_\Psi$. This term is increasing in $p$, which reflects the fact that estimating $\beta^{\mathsf{glm}}$ is more challenging when $p$ is large. As expected, this term is decreasing in $n$ and $|S|$, i.e. larger sample size yields better estimates. When the full OLS solution is used ($|S| = n$), the second term becomes $\mathcal{O}(\sqrt{p\max\{\log(n), p\}/n}) = \mathcal{O}(p/\sqrt{n})$, for $p$ sufficiently large. This suggests that $n$ should be at least of order $p^2$ for good performance.

# 7 Discussion

In this paper, we showed that the coefficients of GLMs and OLS are approximately proportional in the general random design setting. Using this relation, we proposed a computationally efficient algorithm for large-scale problems that achieves the same accuracy as the MLE by first estimating the OLS coefficients and then estimating the proportionality constant through iterations that can attain quadratic or cubic convergence rate, with only $\mathcal{O}(n)$ per-iteration cost.

We briefly mentioned that the proportionality between the coefficients holds even when there is regularization in Section 3.1. Further pursuing this idea may be interesting for large-scale problems where regularization is crucial. Another interesting line of research is to find similar proportionality relations between the parameters in other large-scale optimization problems such as support vector machines. Such relations may reduce the problem complexity significantly.

## Footnotes

*Work conducted while at Rutgers University

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
