[Supplementary Material]

## Supplementary Material

# Scaled Least Squares Estimator for GLMs

We provide all technical details in the Supplementary Material. Section A provides the proofs the main technical results. We provide additional experiments in Section B. In Section C, we state several auxiliary lemmas that are used throughout the proofs.

## A  Proof of Main Results

In this section, we provide the details and the proofs of our technical results. For convenience, we briefly state the following definitions.

**Definition 2** (Sub-Gaussian). *For a given constant $\kappa$, a random variable $x \in \mathbb{R}$ is said to be* sub-Gaussian *if it satisfies*

$$\sup_{m \geq 1} m^{-1/2} \mathbb{E}\left[|x|^m\right]^{1/m} \leq \kappa.$$

*Smallest such $\kappa$ is the sub-Gaussian norm of $x$ and it is denoted by $\|x\|_{\psi_2}$. Similarly, a random vector $y \in \mathbb{R}^p$ is a* sub-Gaussian *vector if there exists a constant $\kappa'$ such that*

$$\sup_{v \in S^{p-1}} \|\langle y, v \rangle\|_{\psi_2} \leq \kappa'.$$

**Definition 3** (Sub-exponential). *For a given constant $\kappa$, a random variable $x \in \mathbb{R}$ is called* sub-exponential *if it satisfies*

$$\sup_{m \geq 1} m^{-1} \mathbb{E}\left[|x|^m\right]^{1/m} \leq \kappa.$$

*Smallest such $\kappa$ is the sub-exponential norm of $x$ and it is denoted by $\|x\|_{\psi_1}$. Similarly, a random vector $y \in \mathbb{R}^p$ is a* sub-exponential *vector if there exists a constant $\kappa'$ such that*

$$\sup_{v \in S^{p-1}} \|\langle y, v \rangle\|_{\psi_1} \leq \kappa'.$$

We start with the proof of Theorem 1.

*Proof of Theorem 1.* For simplicity, we denote the whitened covariate by $w = \mathbf{\Sigma}^{-1/2}x$. Since $w$ is sub-Gaussian with norm $\kappa$, its $j$-th entry $w_j$ has bounded third moment. That is,

$$\begin{aligned} \kappa &= \sup_{\|u\|_2=1} \|\langle u, w \rangle\|_{\psi_2}, \\ &\geq \|w_j\|_{\psi_2} = \sup_{m \geq 1} m^{-1/2} \mathbb{E}\left[|w_j|^m\right]^{1/m}, \\ &\geq \frac{1}{\sqrt{3}} \mathbb{E}\left[|w_j|^3\right]^{1/3}, \end{aligned} \tag{16}$$

where in the first step, we used $u = e_j$, the $j$-th standard basis vector. Hence, we obtain a bound on the third moment, i.e,

$$\max_j \mathbb{E}\left[|w_j|^3\right] \leq 3^{3/2}\kappa^3. \tag{17}$$

Using the normal equations, we write

$$\begin{aligned} \mathbb{E}\left[yx\right] = \mathbb{E}\left[x\Psi^{(1)}(\langle x, \beta \rangle)\right] &= \mathbf{\Sigma}^{1/2}\mathbb{E}\left[w\Psi^{(1)}(\langle w, \mathbf{\Sigma}^{1/2}\beta \rangle)\right], \\ &= \mathbf{\Sigma}^{1/2}\mathbb{E}\left[w\Psi^{(1)}(\langle w, \tilde{\beta} \rangle)\right], \end{aligned} \tag{18}$$

where we defined $\tilde{\beta} = \mathbf{\Sigma}^{1/2}\beta$. By multiplying both sides with $\mathbf{\Sigma}^{-1}$, we obtain

$$\beta^{\mathsf{ols}} = \mathbf{\Sigma}^{-1/2}\mathbb{E}\left[w\Psi^{(1)}(\langle w, \tilde{\beta} \rangle)\right]. \tag{19}$$

Now we define the partial sums $W_{-i} = \sum_{j \neq i} \tilde{\beta}_j w_j = \langle \tilde{\beta}, w \rangle - \tilde{\beta}_i w_i$. We will focus on the $i$-th entry of the above expectation given in (19). Denoting the zero biased transformation of $w_i$ by $w_i^*$, we have

$$
\begin{aligned}
\mathbb{E}\left[w_i \Psi^{(1)}(\langle w, \tilde{\beta} \rangle)\right] &= \mathbb{E}\left[w_i \mathbb{E}\left[\Psi^{(1)}\left(\tilde{\beta}_i w_i + W_{-i}\right) | w_i\right]\right], \\
&= \tilde{\beta}_i \mathbb{E}\left[\Psi^{(2)}(\tilde{\beta}_i w_i^* + W_{-i})\right], \\
&= \tilde{\beta}_i \mathbb{E}\left[\Psi^{(2)}(\tilde{\beta}_i(w_i^* - w_i) + \langle w, \tilde{\beta} \rangle)\right].
\end{aligned}
\tag{20}
$$

Let $\mathbf{D}$ be a diagonal matrix with diagonal entries $\mathbf{D}_{ii} = \mathbb{E}\left[\Psi^{(2)}(\tilde{\beta}_i(w_i^* - w_i) + \langle w, \tilde{\beta} \rangle)\right]$. Using (19) together with (20), we obtain the equality

$$
\begin{aligned}
\beta^{\mathsf{ols}} &= \mathbf{\Sigma}^{-1/2} \mathbf{D} \tilde{\beta}, \\
&= \mathbf{\Sigma}^{-1/2} \mathbf{D} \mathbf{\Sigma}^{1/2} \beta.
\end{aligned}
\tag{21}
$$

Now, using the Lipschitz continuity assumption of the variance function, we have

$$
\left| \mathbb{E}\left[\Psi^{(2)}(\tilde{\beta}_i(w_i^* - w_i) + \langle w, \tilde{\beta} \rangle)\right] - \mathbb{E}\left[\Psi^{(2)}(\langle w, \tilde{\beta} \rangle)\right] \right| \leq k|\tilde{\beta}_i| \mathbb{E}\left[|w_i^* - w_i|\right].
\tag{22}
$$

In the following, we will use the properties of zero-biased transformations. Consider the quantity

$$
r = \sup \frac{\mathbb{E}\left[|w_i^* - w_i|\right]}{\mathbb{E}\left[|w_i|^3\right]}
\tag{23}
$$

where $w_i^*$ has $w_i$-zero biased distribution and the supremum is taken with respect to all random variables with mean 0, standard deviation 1 and finite third moment, and $w_i^*$ is achieving the minimal $\ell_1$ coupling to $w_i$. It is shown in [Gol07] that $\rho$ is upper bounded by 1.5. Then the right hand side of (22) can be upper bounded by

$$
\begin{aligned}
k|\tilde{\beta}_i| \mathbb{E}\left[|w_i^* - w_i|\right] &\leq rk \max_i \left\{|\tilde{\beta}_i| \mathbb{E}\left[|w_i|^3\right]\right\}, \\
&\leq 1.5k \left\|\mathbf{\Sigma}^{1/2}\beta\right\|_\infty 3^{3/2} \kappa^3, \\
&\leq 8k\kappa^3 \|\mathbf{\Sigma}^{1/2}\beta\|_\infty,
\end{aligned}
\tag{24}
$$

where in the second step we used the bound on the third moment given in (17). The last inequality provides us with the following result,

$$
\max_i \left| \mathbf{D}_{ii} - \frac{1}{c_\Psi} \right| \leq 8k\kappa^3 \|\mathbf{\Sigma}^{1/2}\beta\|_\infty.
\tag{25}
$$

Finally, combining this with (19) and (21), we obtain

$$
\begin{aligned}
\left\| \beta^{\mathsf{ols}} - \frac{1}{c_\Psi}\beta \right\|_\infty &= \left\| \mathbf{\Sigma}^{-1/2} \mathbf{D} \mathbf{\Sigma}^{1/2}\beta - \frac{1}{c_\Psi}\beta \right\|_\infty, \\
&= \left\| \mathbf{\Sigma}^{-1/2} \left(\mathbf{D} - \frac{1}{c_\Psi}\mathbf{I}\right) \mathbf{\Sigma}^{1/2}\beta \right\|_\infty, \\
&\leq \max_i \left| \mathbf{D}_{ii} - \frac{1}{c_\Psi} \right| \left\| \mathbf{\Sigma}^{1/2} \right\|_\infty \left\| \mathbf{\Sigma}^{-1/2} \right\|_\infty \|\beta\|_\infty, \\
&\leq 8k\kappa^3 \rho(\mathbf{\Sigma}^{1/2}) \|\mathbf{\Sigma}^{1/2}\|_\infty \frac{\tau^2}{r^2 p},
\end{aligned}
\tag{26}
$$

where in the last step, we used the assumption that $\beta$ is $r$-well-spread. $\qquad\square$

*Proof of Proposition 2.* For convenience, we denote the whitened covariates with $w_i = \mathbf{\Sigma}^{-1/2} x_i$. We have $\mathbb{E}[w_i] = 0$, $\mathbb{E}[w_i w_i^T] = \mathbf{I}$, and $\|w_i\|_{\psi_2} \leq \kappa$. Also denote the sub-sampled covariance

matrix with $\widehat{\boldsymbol{\Sigma}} = \frac{1}{|S|} \sum_{i \in S} x_i x_i^T$, and its whitened version as $\widetilde{\boldsymbol{\Sigma}} = \frac{1}{|S|} \sum_{i \in S} w_i w_i^T$. Further, define $\hat{\zeta} = \frac{1}{n} \sum_{i=1}^{n} w_i y_i$ and $\zeta = \mathbb{E}[wy]$. Then, we have

$$\hat{\beta}^{\mathsf{ols}} = \widehat{\boldsymbol{\Sigma}}^{-1} \boldsymbol{\Sigma}^{1/2} \hat{\zeta} \quad \text{and} \quad \beta^{\mathsf{ols}} = \boldsymbol{\Sigma}^{-1/2} \zeta.$$

For now, we work on the event that $\widehat{\boldsymbol{\Sigma}}$ is invertible. We will see that this event holds with very high probability. We write

$$\left\| \boldsymbol{\Sigma}^{1/2} (\hat{\beta}^{\mathsf{ols}} - \beta^{\mathsf{ols}}) \right\|_2 = \left\| \boldsymbol{\Sigma}^{1/2} \widehat{\boldsymbol{\Sigma}}^{-1} \boldsymbol{\Sigma}^{1/2} \hat{\zeta} - \boldsymbol{\Sigma}^{-1/2} \zeta \right\|_2, \tag{27}$$

$$= \left\| \widetilde{\boldsymbol{\Sigma}}^{-1} \left\{ \hat{\zeta} - \zeta + \left( \mathbf{I} - \boldsymbol{\Sigma}^{-1/2} \widehat{\boldsymbol{\Sigma}} \boldsymbol{\Sigma}^{-1/2} \right) \zeta \right\} \right\|_2,$$

$$\leq \left\| \widetilde{\boldsymbol{\Sigma}}^{-1} \right\|_2 \left\{ \left\| \hat{\zeta} - \zeta \right\|_2 + \left\| \mathbf{I} - \widetilde{\boldsymbol{\Sigma}} \right\|_2 \| \zeta \|_2 \right\},$$

where we used the triangle inequality and the properties of the operator norm.

For the first term on the right hand side of (27), we write

$$\left\| \widetilde{\boldsymbol{\Sigma}}^{-1} \right\|_2 = \frac{1}{\lambda_{\min}(\widetilde{\boldsymbol{\Sigma}})},$$

$$\leq \frac{1}{1 - \delta},$$

where we assumed that such a $\delta > 0$ exists. In fact, when $\delta < 0.5$, we obtain a bound of 2 on the right hand side, which also justifies the invertibility assumption of $\widehat{\boldsymbol{\Sigma}}$. By Lemma 5 and the following remark, we have with probability at least $1 - 2 \exp\{-p\}$,

$$\left\| \widetilde{\boldsymbol{\Sigma}} - \mathbf{I} \right\|_2 \leq c \sqrt{\frac{p}{|S|}},$$

where $c$ is a constant depending only on $\kappa$. When $|S| > 4c^2 p$, we obtain

$$\left| \lambda_{\min}(\widetilde{\boldsymbol{\Sigma}}) - 1 \right| \leq \left\| \widetilde{\boldsymbol{\Sigma}} - \mathbf{I} \right\|_2 \leq 0.5,$$

where the first inequality follows from the Lipschitz property of the eigenvalues.

Next, we bound the difference between $\hat{\zeta}$ and its expectation $\zeta$. We write the bounds on the sub-exponential norm

$$\| wy \|_{\psi_1} = \sup_{\|v\|_2 = 1} \sup_{m \geq 1} m^{-1} \mathbb{E}\left[ |\langle v, w \rangle y|^m \right]^{1/m}, \tag{28}$$

$$\leq \sup_{\|v\|_2 = 1} \sup_{m \geq 1} m^{-1} \mathbb{E}\left[ |\langle v, w \rangle|^{2m} \right]^{1/2m} \mathbb{E}\left[ |y|^{2m} \right]^{1/2m},$$

$$\leq \sup_{\|v\|_2 = 1} \sup_{m \geq 1} m^{-1/2} \mathbb{E}\left[ |\langle v, w \rangle|^{2m} \right]^{1/2m} \sup_{m \geq 1} m^{-1/2} \mathbb{E}\left[ |y|^{2m} \right]^{1/2m},$$

$$\leq 2 \| w \|_{\psi_2} \| y \|_{\psi_2} = 2\gamma\kappa.$$

Hence, we have $\max_i \| w_i y_i - \mathbb{E}[w_i y_i] \|_{\psi_1} \leq 4\gamma\kappa$. Further, let $e_j$ denote the $j$-th standard basis, and notice that each entry of $w$ is also sub-Gaussian with norm upper bounded by $\kappa$, i.e.,

$$\kappa = \| w \|_{\psi_2} = \sup_{\|u\|_2 = 1} \| \langle u, w \rangle \|_{\psi_2}, \tag{29}$$

$$\geq \| \langle e_j, w \rangle \|_{\psi_2} = \| w_j \|_{\psi_2}.$$

Also, we can write

$$2\gamma\kappa \geq \| wy \|_{\psi_1} = \sup_{\|u\|_2 = 1} \sup_{m \geq 1} m^{-1} \mathbb{E}\left[ |\langle u, w \rangle y|^m \right]^{1/m}, \tag{30}$$

$$\geq \sup_{\|u\|_2 = 1} \mathbb{E}\left[ |\langle u, w \rangle y| \right],$$

$$\geq \sup_{\|u\|_2 = 1} \mathbb{E}\left[ \langle u, w \rangle y \right],$$

$$= \sup_{\|u\|_2 = 1} \langle u, \zeta \rangle = \| \zeta \|_2,$$

where in the last step, we used the fact that dual norm of $\ell_2$ norm is itself.

Next, we apply Lemma 2 to $\hat{\zeta} - \zeta$, and obtain with probability at least $1 - \exp\{-p\}$

$$\left\| \hat{\zeta} - \zeta \right\|_2 \le c\gamma\kappa \sqrt{\frac{p}{n}},$$

whenever $n > c^2 p$ for an absolute constant $c$.

Combining the above results in (27), we obtain with probability at least $1 - 3\exp\{-p\}$

$$\left\| \mathbf{\Sigma}^{1/2}(\hat{\beta}^{\mathsf{ols}} - \beta^{\mathsf{ols}}) \right\|_2 \le 2\left\{ c_1 \gamma\kappa \sqrt{\frac{p}{n}} + c_2 \gamma\kappa \sqrt{\frac{p}{|S|}} \right\} \le \eta \sqrt{\frac{p}{|S|}} \tag{31}$$

where $\eta$ depends only on $\kappa$ and $\gamma$, and $|S| > \eta p$. Finally, we write

$$\left\| \hat{\beta}^{\mathsf{ols}} - \beta^{\mathsf{ols}} \right\|_2 \le \lambda_{\min}^{-1/2} \left\| \mathbf{\Sigma}^{1/2}(\hat{\beta}^{\mathsf{ols}} - \beta^{\mathsf{ols}}) \right\|_2,$$

$$\le \eta \lambda_{\min}^{-1/2} \sqrt{\frac{p}{|S|}},$$

with probability at least $1 - 3\exp\{-p\}$, whenever $|S| > \eta p$. $\qquad\square$

The following lemma – combined with the Proposition 2 – provides the necessary tools to prove Theorem 2.

**Lemma 1.** *For a given function $\Psi^{(2)}$ that is Lipschitz continuous with $k$, and uniformly bounded by $b$, we define the function $f : \mathbb{R} \times \mathbb{R}^p \to \mathbb{R}$ as*

$$f(c, \beta) = c\, \mathbb{E}\left[ \Psi^{(2)}(\langle x, \beta \rangle c) \right],$$

*and its empirical counterpart as*

$$\hat{f}(c, \beta) = c\, \frac{1}{n} \sum_{i=1}^{n} \Psi^{(2)}(\langle x_i, \beta \rangle c).$$

*Assume that for some $\delta, \bar{c} > 0$, we have $f(\bar{c}, \beta^{\mathsf{ols}}) \ge 1 + \delta$. Then, $\exists c_\Psi > 0$ satisfying the equation*

$$1 = f(c_\Psi, \beta^{\mathsf{ols}}).$$

*Further, assume that for some $\tilde{\delta} > 0$, we have $\delta = \tilde{\delta}\sqrt{p}$, and $n$ and $|S|$ sufficiently large, i.e.,*

$$\min\left\{ \frac{n}{\log(n)}, \frac{|S|}{p} \right\} > K^2/\tilde{\delta}^2$$

*for $K = \eta\bar{c}\max\{b + \kappa/\tilde{\mu}, k\bar{c}\}$. Then, with probability $1 - 5\exp\{-p\}$, there exists a constant $\hat{c}_\Psi \in (0, \bar{c})$ satisfying the equation*

$$1 = \hat{c}_\Psi \frac{1}{n} \sum_{i=1}^{n} \Psi^{(2)}(\langle x_i, \hat{\beta}^{\mathsf{ols}} \rangle \hat{c}_\Psi).$$

*Moreover, if the derivative of $z \to f(z, \beta^{\mathsf{ols}})$ is bounded below in absolute value (i.e. does not change sign) by $\upsilon > 0$ in the interval $z \in [0, \bar{c}]$, then with probability $1 - 5\exp\{-p\}$, we have*

$$|\hat{c}_\Psi - c_\Psi| \le C \sqrt{\frac{p}{\min\{n/\log(n), |S|/p\}}},$$

*where $C = K/\upsilon$.*

*Proof of Lemma 1.* First statement is obvious. We notice that $f(c, \beta^{\mathsf{ols}})$ is a continuous function in its first argument with $f(0, \beta^{\mathsf{ols}}) = 0$ and $f(\bar{c}, \beta^{\mathsf{ols}}) \ge 1 + \delta$. Hence, there exists $c_\Psi > 0$ such that $f(c_\Psi, \beta^{\mathsf{ols}}) = 1$. If there are many solutions to the above equation, we choose the one that is closest to zero. The condition on the derivative will guarantee the uniqueness of the solution.

Next, we will show the existence of $\hat{c}_\Psi$ using a uniform concentration given by Lemma 3. Define the ellipsoid centered around $\beta^{\mathsf{ols}}$ with radius $\delta$,

$$\mathcal{B}_{\boldsymbol{\Sigma}}^\delta(\beta^{\mathsf{ols}}) = \left\{\beta : \left\|\boldsymbol{\Sigma}^{1/2}(\beta - \beta^{\mathsf{ols}})\right\|_2 \le \delta\right\},$$

and the event $\mathcal{E}$ that $\hat{\beta}^{\mathsf{ols}}$ falls into $\mathcal{B}_{\boldsymbol{\Sigma}}^\delta(\beta^{\mathsf{ols}})$, i.e.,

$$\mathcal{E} = \left\{\hat{\beta}^{\mathsf{ols}} \in \mathcal{B}_{\boldsymbol{\Sigma}}^\delta(\beta^{\mathsf{ols}})\right\}.$$

By Proposition 2 and the inequality given in (31), whenever $|S| > \eta p \max\left\{1, \eta/\delta^2\right\}$, we obtain

$$\mathbb{P}\left(\mathcal{E}^C\right) \le 3\exp\left\{-p\right\},$$

where $\mathcal{E}^C$ denotes the complement of the event $\mathcal{E}$, and $\eta$ is a constant depending only on $\kappa$ and $\gamma$. For any $c \in [0, \bar{c}]$, on the event $\mathcal{E}$, we have

$$\left|\hat{f}(c, \hat{\beta}^{\mathsf{ols}}) - f(c, \hat{\beta}^{\mathsf{ols}})\right| \le \sup_{\beta \in \mathcal{B}_{\boldsymbol{\Sigma}}^\delta(\beta^{\mathsf{ols}})} \left|\hat{f}(c, \beta) - f(c, \beta)\right|.$$

Hence, we obtain the following inequality

$$\mathbb{P}\left(\sup_{c \in [0,\bar{c}]} \left|\hat{f}(c, \hat{\beta}^{\mathsf{ols}}) - f(c, \hat{\beta}^{\mathsf{ols}})\right| > \epsilon\right) \le \mathbb{P}\left(\sup_{c \in [0,\bar{c}]} \left|\hat{f}(c, \hat{\beta}^{\mathsf{ols}}) - f(c, \hat{\beta}^{\mathsf{ols}})\right| > \epsilon; \mathcal{E}\right) + \mathbb{P}\left(\mathcal{E}^C\right),$$

$$\le \mathbb{P}\left(\sup_{c \in [0,\bar{c}]} \sup_{\beta \in \mathcal{B}_{\boldsymbol{\Sigma}}^\delta(\beta^{\mathsf{ols}})} \left|\hat{f}(c, \beta) - f(c, \beta)\right| > \epsilon\right) + 3\exp\left\{-p\right\}.$$

In the following, we will use Lemma 3 for the first term in the last line above. Denoting by $w$, the whitened covariates, we have $\langle x, \beta \rangle = \langle w, \boldsymbol{\Sigma}^{1/2}\beta \rangle$. Therefore,

$$\sup_{c \in [0,\bar{c}]} \sup_{\beta \in \mathcal{B}_{\boldsymbol{\Sigma}}^\delta(\beta^{\mathsf{ols}})} \left|\hat{f}(c, \beta) - f(c, \beta)\right|$$

$$\le \bar{c} \sup_{c \in [0,\bar{c}]} \sup_{\beta \in \mathcal{B}_{\boldsymbol{\Sigma}}^\delta(\beta^{\mathsf{ols}})} \left|\frac{1}{n}\sum_{i=1}^n \Psi^{(2)}(\langle w_i, \boldsymbol{\Sigma}^{1/2}\beta \rangle c) - \mathbb{E}\left[\Psi^{(2)}(\langle w, \boldsymbol{\Sigma}^{1/2}\beta \rangle c)\right]\right|.$$

Next, define the ball centered around $\tilde{\beta}^{\mathsf{ols}} = \boldsymbol{\Sigma}^{1/2}\beta^{\mathsf{ols}}$, with radius $\delta$ as $\mathcal{B}_\delta(\tilde{\beta}^{\mathsf{ols}}) = \boldsymbol{\Sigma}^{1/2}\mathcal{B}_{\boldsymbol{\Sigma}}^\delta(\beta^{\mathsf{ols}})$. We have $\beta \in \mathcal{B}_{\boldsymbol{\Sigma}}^\delta(\beta^{\mathsf{ols}})$ if and only if $\boldsymbol{\Sigma}^{1/2}\beta \in \mathcal{B}_\delta(\tilde{\beta}^{\mathsf{ols}})$. Then, the right hand side of the above inequality can be written as

$$\bar{c} \sup_{c \in [0,\bar{c}]} \sup_{\beta \in \mathcal{B}_\delta(\tilde{\beta}^{\mathsf{ols}})} \left|\frac{1}{n}\sum_{i=1}^n \Psi^{(2)}(\langle w_i, \beta \rangle c) - \mathbb{E}\left[\Psi^{(2)}(\langle w, \beta \rangle c)\right]\right|,$$

$$= \bar{c} \sup_{\beta \in \mathcal{B}_{\bar{c}\delta}(\tilde{\beta}^{\mathsf{ols}})} \left|\frac{1}{n}\sum_{i=1}^n \Psi^{(2)}(\langle w_i, \beta \rangle) - \mathbb{E}\left[\Psi^{(2)}(\langle w, \beta \rangle)\right]\right|.$$

Then, by Lemma 3, we obtain

$$\mathbb{P}\left(\sup_{c \in [0,\bar{c}]} \left|\hat{f}(c, \hat{\beta}^{\mathsf{ols}}) - f(c, \hat{\beta}^{\mathsf{ols}})\right| > c'\bar{c}(b + \kappa/\tilde{\mu})\sqrt{\frac{p}{n/\log(n)}}\right) \le 5\exp\left\{-p\right\} \qquad (32)$$

whenever $np > 51\max\left\{\chi, \chi^{-1}\right\}$ where $\chi = (b + \kappa/\tilde{\mu})^2/(c'\delta^2 k^2 \bar{c}^2 \tilde{\mu}^2)$.

Also, by the Lipschitz condition for $\Psi^{(2)}$, we have for any $c \in [0, \bar{c}]$, and $\beta_1, \beta_2$,

$$|f(c, \beta_1) - f(c, \beta_2)| \le kc^2 \mathbb{E}\left[\left|\langle w, \boldsymbol{\Sigma}^{1/2}(\beta_1 - \beta_2) \rangle\right|\right]$$

$$\le k\bar{c}^2 \mathbb{E}\left[\|w\|_2\right]\left\|\boldsymbol{\Sigma}^{1/2}(\beta_1 - \beta_2)\right\|_2$$

$$\le k\bar{c}^2 \sqrt{p}\left\|\boldsymbol{\Sigma}^{1/2}(\beta_1 - \beta_2)\right\|_2$$

Applying the above bound for $\beta_1 = \hat{\beta}^{\mathsf{ols}}$ and $\beta_2 = \beta^{\mathsf{ols}}$, we obtain with probability $1 - 3\exp\{-p\}$

$$\left| f(c, \hat{\beta}^{\mathsf{ols}}) - f(c, \beta^{\mathsf{ols}}) \right| \le \eta k\bar{c}^2 \frac{p}{\sqrt{|S|}}, \tag{33}$$

where the last step follows from Proposition 2 and the inequality given in (31).

Combining this with the previous bound, and taking into account that $\mu = \tilde{\mu}\sqrt{p}$, for any $c \in [0, \bar{c}]$, with probability $1 - 5\exp\{-p\}$, we obtain

$$\left| \hat{f}(c, \hat{\beta}^{\mathsf{ols}}) - f(c, \beta^{\mathsf{ols}}) \right| \le c'\bar{c}(b + \kappa/\tilde{\mu})\sqrt{\frac{p}{n/\log(n)}} + \eta k\bar{c}^2 \frac{p}{\sqrt{|S|}}$$

$$\le K\sqrt{\frac{p}{\min\{n/\log(n), |S|/p\}}}$$

where $K = \eta\bar{c}\max\{b + \kappa/\tilde{\mu}, k\bar{c}\}$. Here, $\eta$ depends only on $\kappa$ and $\gamma$.

In particular, for $c = \bar{c}$ we observe that

$$\hat{f}(\bar{c}, \hat{\beta}^{\mathsf{ols}}) \ge f(\bar{c}, \beta^{\mathsf{ols}}) - K\sqrt{\frac{p}{\min\{n/\log(n), |S|/p\}}}$$

$$\ge 1 + \delta - K\sqrt{\frac{p}{\min\{n/\log(n), |S|/p\}}}.$$

Therefore, for sufficiently large $n$ and $|S|$ satisfying

$$\min\left\{\frac{n}{\log(n)}, \frac{|S|}{p}\right\} > K^2/\tilde{\delta}^2$$

we obtain $\hat{f}(\bar{c}, \hat{\beta}^{\mathsf{ols}}) > 1$. Since this function is continuous and $\hat{f}(0, \hat{\beta}^{\mathsf{ols}}) = 0$, we obtain the existence of $\hat{c}_\Psi \in [0, \bar{c}]$ with probability at least $1 - 5\exp\{-p\}$.

Now, since $\hat{c}_\Psi$ and $c_\Psi$ satisfy the equations $\hat{f}(\hat{c}_\Psi, \hat{\beta}^{\mathsf{ols}}) = f(c_\Psi, \beta^{\mathsf{ols}}) = 1$ (with high probability), by the inequality given in (32), with probability at least $1 - 5\exp\{-p\}$, we obtain

$$\left| 1 - f(\hat{c}_\Psi, \hat{\beta}^{\mathsf{ols}}) \right| = \left| \hat{f}(\hat{c}_\Psi, \hat{\beta}^{\mathsf{ols}}) - f(\hat{c}_\Psi, \hat{\beta}^{\mathsf{ols}}) \right|$$

$$\le c'\bar{c}(b + \kappa/\tilde{\mu})\sqrt{\frac{p}{n/\log(n)}}.$$

Also, by the same argument in (33), and Proposition 2, we get

$$\left| f(\hat{c}_\Psi, \hat{\beta}^{\mathsf{ols}}) - f(\hat{c}_\Psi, \beta^{\mathsf{ols}}) \right| \le k\bar{c}^2 \sqrt{p} \left\| \mathbf{\Sigma}(\hat{\beta}^{\mathsf{ols}} - \beta^{\mathsf{ols}}) \right\|_2$$

$$\le \eta k\bar{c}^2 \frac{p}{\sqrt{|S|}}.$$

Now, using the Taylor's series expansion of $c \to f(c, \beta^{\mathsf{ols}})$ around $c_\Psi$, and the assumption on the derivative of $f$ with respect to its first argument, we obtain

$$\upsilon \left| \hat{c}_\Psi - c_\Psi \right| \le \left| f(\hat{c}_\Psi, \beta^{\mathsf{ols}}) - f(c_\Psi, \beta^{\mathsf{ols}}) \right|$$

$$\le \left| f(\hat{c}_\Psi, \beta^{\mathsf{ols}}) - f(\hat{c}_\Psi, \hat{\beta}^{\mathsf{ols}}) \right| + \left| f(\hat{c}_\Psi, \hat{\beta}^{\mathsf{ols}}) - 1 \right|$$

$$\le \eta k\bar{c}^2 \frac{p}{\sqrt{|S|}} + c'\bar{c}(b + \kappa/\tilde{\mu})\sqrt{\frac{p}{n/\log(n)}}$$

$$\le K\sqrt{\frac{p}{\min\{n/\log(n), |S|/p\}}}$$

with probability at least $1 - 5\exp\{-p\}$. Here, the constant $K$ is the same as before

$$K = \eta\bar{c}\max\{b + \kappa/\tilde{\mu}, k\bar{c}\}.$$

$\square$

*Proof of Theorem 2.* We have

$$\left\|\hat{\beta}^{\mathsf{sls}} - \beta^{\mathsf{glm}}\right\|_\infty = \left\|\hat{c}_\Psi \hat{\beta}^{\mathsf{ols}} - \beta^{\mathsf{glm}}\right\|_\infty, \tag{34}$$

$$\leq \left\|c_\Psi \beta^{\mathsf{ols}} - \beta^{\mathsf{glm}}\right\|_\infty + \left\|\hat{c}_\Psi \hat{\beta}^{\mathsf{ols}} - c_\Psi \beta^{\mathsf{ols}}\right\|_\infty,$$

where we used the triangle inequality for the $\ell_\infty$ norm. The first term on the right hand side can be bounded using Theorem 1. We write

$$\left\|c_\Psi \beta^{\mathsf{ols}} - \beta^{\mathsf{glm}}\right\|_\infty \leq \eta_1 \frac{1}{p}, \tag{35}$$

for $\eta_1 = 8k\bar{c}\kappa^3 \rho(\mathbf{\Sigma}^{1/2})\|\mathbf{\Sigma}^{1/2}\|_\infty (\tau/r)^2$.

For the second term, we write

$$\left\|\hat{c}_\Psi \hat{\beta}^{\mathsf{ols}} - c_\Psi \beta^{\mathsf{ols}}\right\|_\infty = \left\|\hat{c}_\Psi \hat{\beta}^{\mathsf{ols}} \pm \hat{c}_\Psi \beta^{\mathsf{ols}} - c_\Psi \beta^{\mathsf{ols}}\right\|_\infty, \tag{36}$$

$$\leq \left\|\hat{c}_\Psi \hat{\beta}^{\mathsf{ols}} - \hat{c}_\Psi \beta^{\mathsf{ols}}\right\|_\infty + \left\|\hat{c}_\Psi \beta^{\mathsf{ols}} - c_\Psi \beta^{\mathsf{ols}}\right\|_\infty,$$

$$\leq |\hat{c}_\Psi| \left\|\hat{\beta}^{\mathsf{ols}} - \beta^{\mathsf{ols}}\right\|_\infty + |\hat{c}_\Psi - c_\Psi| \left\|\beta^{\mathsf{ols}}\right\|_\infty,$$

where the first step follows from triangle inequality. By Lemma 1, for sufficiently large $n$ and $|S|$, with probability $1 - 5\exp\{-p\}$, the constant $\hat{c}_\Psi$ exists and it is in the interval $(0, \bar{c}]$. By the same lemma, with probability $1 - 5\exp\{-p\}$, we have

$$|\hat{c}_\Psi - c_\Psi| \leq \eta_4 \sqrt{\frac{p}{\min\{n/\log(n), |S|/p\}}}, \tag{37}$$

where $\eta_4 = \eta' \upsilon^{-1} \bar{c} \max\{b + \kappa/\tilde{\mu}, k\bar{c}\}$, for some constant $\eta'$ depending on the sub-Gaussian norms $\kappa$ and $\gamma$.

Also, by the norm equivalence and Proposition 2, we have with probability $1 - 3\exp\{-p\}$

$$\left\|\hat{\beta}^{\mathsf{ols}} - \beta^{\mathsf{ols}}\right\|_\infty \leq \eta_3 \sqrt{\frac{p}{|S|}}, \tag{38}$$

for $\eta_3 = \eta'' \lambda_{\min}^{-1/2}$, where $\eta''$ is constant depending only on $\gamma$ and $\kappa$.

Finally, combining all these inequalities with the last line of (34), we have with probability $1 - 5\exp\{-p\}$,

$$\left\|\hat{\beta}^{\mathsf{sls}} - \beta^{\mathsf{glm}}\right\|_\infty \leq \eta_1 \frac{1}{p} + \eta_3 \bar{c} \sqrt{\frac{p}{|S|}} + \eta_4 \left\|\beta^{\mathsf{ols}}\right\|_\infty \sqrt{\frac{p}{\min\{n/\log(n), |S|/p\}}}, \tag{39}$$

$$\leq \eta_1 \frac{1}{p} + \left(\eta_3 \bar{c} + \eta_4 \left\|\beta^{\mathsf{ols}}\right\|_\infty\right) \sqrt{\frac{p}{\min\{n/\log(n), |S|/p\}}},$$

$$= \eta_1 \frac{1}{p} + \eta_2 \sqrt{\frac{p}{\min\{n/\log(n), |S|/p\}}},$$

where

$$\eta_1 = 8k\bar{c}\kappa^3 \rho(\mathbf{\Sigma}^{1/2})\|\mathbf{\Sigma}^{1/2}\|_\infty (\tau/r)^2 \tag{40}$$

$$\eta_2 = \eta_3 \bar{c} + \eta_4 \left\|\beta^{\mathsf{ols}}\right\|_\infty,$$

$$= \eta \bar{c} \lambda_{\min}^{-1/2} \left(1 + \upsilon^{-1} \lambda_{\min}^{1/2}\|\beta^{\mathsf{ols}}\|_\infty \max\{(b + k/\tilde{\mu}), k\bar{c}\}\right).$$

$\square$

*Proof of Corollary 1.* The normal equations for the lasso minimization yields

$$\mathbb{E}\left[xx^T\right] \beta_\lambda^{\mathsf{lasso}} - \beta^{\mathsf{ols}} + \lambda s = 0,$$

where $s \in \partial \left\| \beta_\lambda^{\mathsf{lasso}} \right\|_1$. It is well-known that under the orthogonal design where the covariates have i.i.d. entries, the above equation reduces to

$$\mathsf{soft}(\beta^{\mathsf{ols}}; \lambda) = \beta_\lambda^{\mathsf{lasso}},$$

where $\mathsf{soft}(\,\cdot\,; \lambda)$ denotes the soft thresholding operator at level $\lambda$. For any $\beta \in \mathbb{R}^p$, let $\mathrm{supp}(\beta)$ denote the support of $\beta$, i.e., the set $\{i \in [p] : \beta_i \neq 0\}$. We have

$$\mathrm{supp}(\beta_\lambda^{\mathsf{lasso}}) = \{i \in [p] : \beta_{\lambda,i}^{\mathsf{lasso}} \neq 0\},$$
$$= \{i \in [p] : |\beta_i^{\mathsf{ols}}| > \lambda\}$$

By Theorem 1, we have

$$|\beta_i^{\mathsf{ols}}| \leq \frac{1}{c_\Psi} |\beta_i^{\mathsf{glm}}| + \frac{\eta}{|\mathrm{supp}(\beta^{\mathsf{glm}})|},$$

which implies that

$$\mathrm{supp}(\beta_\lambda^{\mathsf{lasso}}) \subset \left\{ i \in [p] : \frac{1}{c_\Psi} |\beta_i^{\mathsf{glm}}| + \frac{\eta}{|\mathrm{supp}(\beta^{\mathsf{glm}})|} > \lambda \right\}.$$

Hence, whenever $\lambda > \eta / |\mathrm{supp}(\beta^{\mathsf{glm}})|$, we have

$$\mathrm{supp}(\beta_\lambda^{\mathsf{lasso}}) \subset \mathrm{supp}(\beta^{\mathsf{glm}}).$$

Further, we have by Theorem 1

$$\frac{1}{c_\Psi} |\beta_i^{\mathsf{glm}}| \leq |\beta_i^{\mathsf{ols}}| + \frac{\eta}{|\mathrm{supp}(\beta^{\mathsf{glm}})|}.$$

Hence, whenever $|\beta_i^{\mathsf{glm}}| > c_\Psi \left( \lambda + \eta / |\mathrm{supp}(\beta^{\mathsf{glm}})| \right)$, we get $|\beta_i^{\mathsf{ols}}| > \lambda$. If this condition is satisfied for any entry in the support of $\beta^{\mathsf{glm}}$, the corresponding lasso coefficient will be non-zero. Therefore, we get

$$\mathrm{supp}(\beta^{\mathsf{glm}}) \subset \mathrm{supp}(\beta_\lambda^{\mathsf{lasso}})$$

under this assumption. Combining this with the previous result, we conclude the proof. $\qquad\square$

# B  Additional Experiments

In this section, we provide additional experiments. The setting is the same as in Section 5. The only difference is the sampling distribution of the datasets, which are stated in the title of each plot. As in Section 5, SLS estimator outperforms its competitors by a large margin in terms of the computation time.

Figure 3: Additional experiments comparing the performance of SLS to that of MLE obtained with various optimization algorithms on several datasets. SLS is represented with red straight line. The details are provided in Table 2

Table 2: Details of the experiments shown in Figure 3.

| MODEL | LOGISTIC REGRESSION | | | | POISSON REGRESSION | | | |
|---|---|---|---|---|---|---|---|---|
| DATASET | $\Sigma \times \text{BER}(\pm 1)$ | | $\Sigma \times \text{NORM}(0,1)$ | | $\Sigma \times \{\text{EXP}(1)\text{-}1\}$ | | $\Sigma \times \text{NORM}(0,1)$ | |
| SIZE | $n = 6.0 \times 10^5, p = 300$ | | $n = 6.0 \times 10^5, p = 300$ | | $n = 6.0 \times 10^5, p = 300$ | | $n = 6.0 \times 10^5, p = 300$ | |
| INITIALIZE | RND | OLS | RND | OLS | RND | OLS | RND | OLS |
| PLOT | (A) | (B) | (C) | (D) | (E) | (F) | (G) | (H) |
| METHOD↓ | TIME IN SECONDS / NUMBER OF ITERATIONS (TO REACH MIN TEST ERROR) | | | | | | | |
| SLS | 6.61/3 | 2.97/3 | 9.38/5 | 4.25/4 | 14.68/4 | 2.99/4 | 6.66/10 | 4.13/10 |
| NR | 222.21/6 | 84.08/3 | 186.33/6 | 115.76/4 | 218.1/6 | 218.9/4 | 364.63/9 | 363.4/9 |
| NS | 40.68/10 | 11.57/3 | 53.06/9 | 19.52/4 | 39.22/6 | 59.61/4 | 51.48/10 | 39.8/10 |
| BFGS | 125.83/33 | 35.41/9 | 155.3/48 | 24.78/8 | 46.61/20 | 48.71/12 | 92.84/36 | 74.22/38 |
| LBFGS | 142.09/38 | 44.41/12 | 444.62/143 | 21.79/7 | 96.53/39 | 50.56/12 | 296.4/111 | 228.1/117 |
| GD | 409.9/134 | 79.45/22 | 1773.1/509 | 135.62/44 | 569.1/211 | 124.31/48 | 792.3/344 | 1041.1/366 |
| AGD | 177.3/159 | 43.76/12 | 359.56/95 | 53.73/18 | 157.9/57 | 63.16/16 | 74.74/32 | 62.21/32 |

## C   Auxiliary Lemmas

**Lemma 2** (Sub-exponential vector concentration). *Let $x_1, x_2, ..., x_n$ be independent centered sub-exponential random vectors with $\max_i \|x_i\|_{\psi_1} = \kappa$. Then we have*

$$\mathbb{P}\left( \left\| \frac{1}{n} \sum_{i=1}^{n} x_i \right\|_2 > c\kappa\sqrt{\frac{p}{n}} \right) \leq \exp\{-p\}. \tag{41}$$

*whenever $n > 4c^2 p$ for an absolute constant $c$.*

*Proof of Lemma 2.* For a vector $z \in \mathbb{R}^p$, we have $\|z\|_2 = \sup_{\|u\|_2=1} \langle u, z \rangle$ since the dual of $\ell_2$ norm is itself. Therefore, we write

$$\mathbb{P}\left( \left\| \frac{1}{n} \sum_{i=1}^{n} x_i \right\|_2 > t \right) = \mathbb{P}\left( \sup_{\|u\|_2=1} \frac{1}{n} \sum_{i=1}^{n} \langle u, x_i \rangle > t \right).$$

Now, let $\mathcal{N}_\epsilon$ be an $\epsilon$-net over $\mathcal{S}^{p-1} = \{u \in \mathbb{R}^p : \|u\|_2 = 1\}$, and observe that

$$\max_{u \in \mathcal{N}_\epsilon} \langle u, x \rangle \geq (1-\epsilon) \sup_{\|u\|_2=1} \langle u, x \rangle,$$
$$= (1-\epsilon)\|x\|_2,$$

with $|\mathcal{N}_\epsilon| \leq (1 + 2/\epsilon)^p$. Hence, we may write

$$\mathbb{P}\left( \sup_{\|u\|_2=1} \frac{1}{n} \sum_{i=1}^{n} \langle u, x_i \rangle > t \right) \leq \mathbb{P}\left( \max_{u \in \mathcal{N}_\epsilon} \frac{1}{n} \sum_{i=1}^{n} \langle u, x_i \rangle > t(1-\epsilon) \right),$$
$$\leq |\mathcal{N}_\epsilon| \mathbb{P}\left( \frac{1}{n} \sum_{i=1}^{n} \langle u, x_i \rangle > t(1-\epsilon) \right).$$

For any $u \in \mathcal{S}^{p-1}$, we have $\|\langle u, x_i \rangle\|_{\psi_1} \leq \kappa$. Then, by the Bernstein-type inequality for sub-exponential random variables [Ver10], we have

$$\mathbb{P}\left( \frac{1}{n} \sum_{i=1}^{n} \langle u, x_i \rangle > t(1-\epsilon) \right) \leq \exp\left\{ -cn \min\left\{ \frac{t^2(1-\epsilon)^2}{\kappa^2}, \frac{t(1-\epsilon)}{\kappa} \right\} \right\},$$

for an absolute constant $c$. Therefore, the probability on the left hand side of (41) can be bounded by

$$\left( 1 + \frac{2}{\epsilon} \right)^p \exp\left\{ -cn\frac{t^2(1-\epsilon)^2}{\kappa^2} \right\} = \exp\left\{ -cn\frac{t^2(1-\epsilon)^2}{\kappa^2} + p\log\left( 1 + \frac{2}{\epsilon} \right) \right\},$$

whenever $t < \kappa/(1-\epsilon)$. Choosing $\epsilon = 0.5$ and for an absolute constant $c' > 3.24/c$ and letting

$$t = c'\kappa\sqrt{\frac{p}{n}},$$

we conclude the proof. $\qquad\qquad\square$

**Lemma 3.** *Let $B(\tilde{\beta})$ denote the ball centered around $\tilde{\beta}$ with radius $\delta$, i.e.,*

$$B(\tilde{\beta}) = \left\{ \beta : \left\| \beta - \tilde{\beta} \right\|_2 \leq \delta \right\}.$$

*For $i = 1, ..., n$, let $x_i \in \mathbb{R}^p$ be i.i.d. centered sub-Gaussian random vectors with norm bounded by $\kappa$ and $\mathbb{E}\left[ \|x\|_2 \right] = \tilde{\mu}\sqrt{p}$. Given a function $g : \mathbb{R} \to \mathbb{R}$ that is uniformly bounded by $b > 0$, and Lipschitz continuous with $k$,*

$$\mathbb{P}\left( \sup_{\beta \in B} \left| \frac{1}{n} \sum_{i=1}^{n} g(\langle x_i, \beta \rangle) - \mathbb{E}\left[ g(\langle x, \beta \rangle) \right] \right| > c(b + \kappa/\tilde{\mu})\sqrt{\frac{p}{n/\log(n)}} \right) \leq 2\exp\{-p\},$$

*whenever $np > 51\max\{\chi, \chi^{-1}\}$ for $\chi = (b + \kappa/\tilde{\mu})^2/(c\delta^2 k^2 \tilde{\mu}^2)$. Above, $c$ is an absolute constant.*

*Proof of Lemma 3.* Let $\mathbb{E}\left[\|x\|_2\right] = \mu = \tilde{\mu}\sqrt{p}$ and for $\epsilon > 0$, $\beta \in B(\tilde{\beta})$ and $w \in \mathbb{R}^p$ define the bounding functions

$$
\begin{aligned}
l_\beta(w) &= g(\langle w, \beta \rangle) - \epsilon \|w\|_2/4\mu, \\
u_\beta(w) &= g(\langle w, \beta \rangle) + \epsilon \|w\|_2/4\mu.
\end{aligned}
$$

Let $\mathcal{N}_\Delta$ be a net over $B(\tilde{\beta})$ in the sense that for any $\beta_1 \in B(\tilde{\beta})$, $\exists \beta_2 \in \mathcal{N}_\Delta$ such that $\|\beta_1 - \beta_2\|_2 \leq \Delta$. We fix $\Delta_* = \epsilon/(4k\mu)$ and write $\forall \beta_1 \in B$, $\exists \beta_2 \in \mathcal{N}_{\Delta_*}$,

1. an upper bound of the form:

$$
\begin{aligned}
g(\langle w, \beta_1 \rangle) &\leq g(\langle w, \beta_2 \rangle) + k\left|\langle w, \beta_1 - \beta_2 \rangle\right|, \\
&\leq g(\langle w, \beta_2 \rangle) + k\|w\|_2 \Delta_*, \\
&= u_{\beta_2}(w),
\end{aligned}
$$

2. and a lower bound of the form:

$$
\begin{aligned}
g(\langle w, \beta_1 \rangle) &\geq g(\langle w, \beta_2 \rangle) - k\left|\langle w, \beta_1 - \beta_2 \rangle\right|, \\
&\geq g(\langle w, \beta_2 \rangle) - k\|w\|_2 \Delta_*, \\
&= l_{\beta_2}(w),
\end{aligned}
$$

where the second steps in the above inequalities follow from the Cauchy-Schwarz inequality. These functions are called *bracketing functions* in the context of empirical process theory.

Hence, we can write that $\forall \beta_1 \in B(\tilde{\beta})$, $\exists \beta_2 \in \mathcal{N}_{\Delta_*}$ such that

$$
\frac{1}{n}\sum_{i=1}^n l_{\beta_2}(x_i) - \mathbb{E}\left[l_{\beta_2}(x)\right] - \epsilon/2 \leq \frac{1}{n}\sum_{i=1}^n g(\langle x_i, \beta_1 \rangle) - \mathbb{E}\left[g(\langle x, \beta_1 \rangle)\right],
$$

$$
\leq \frac{1}{n}\sum_{i=1}^n u_{\beta_2}(x_i) - \mathbb{E}\left[u_{\beta_2}(x)\right] + \epsilon/2.
$$

The above inequalities translate to the following conclusion: Whenever the following event happens,

$$
\left\{ \left| \frac{1}{n}\sum_{i=1}^n g(\langle x_i, \beta_1 \rangle) - \mathbb{E}\left[g(\langle x, \beta_1 \rangle)\right] \right| > \epsilon \right\},
$$

at least one of the following events happens

$$
\left\{ \frac{1}{n}\sum_{i=1}^n u_{\beta_2}(x_i) - \mathbb{E}\left[u_{\beta_2}(x)\right] > \epsilon/2 \right\} \quad \text{or} \quad \left\{ \frac{1}{n}\sum_{i=1}^n l_{\beta_2}(x_i) - \mathbb{E}\left[l_{\beta_2}(x)\right] < -\epsilon/2 \right\}.
$$

Therefore, using the union bound on the above events, we may obtain

$$
\mathbb{P}\left( \sup_{\beta \in B(\tilde{\beta})} \left| \frac{1}{n}\sum_{i=1}^n g(\langle x_i, \beta \rangle) - \mathbb{E}\left[g(\langle x, \beta \rangle)\right] \right| > \epsilon \right) \tag{42}
$$

$$
\leq \mathbb{P}\left( \max_{\beta \in \mathcal{N}_{\Delta_*}} \frac{1}{n}\sum_{i=1}^n u_\beta(x_i) - \mathbb{E}\left[u_\beta(x)\right] > \epsilon/2 \right)
$$

$$
+ \mathbb{P}\left( \max_{\beta \in \mathcal{N}_{\Delta_*}} \frac{1}{n}\sum_{i=1}^n l_\beta(x_i) - \mathbb{E}\left[l_\beta(x)\right] < -\epsilon/2 \right).
$$

Note that the right hand side of the above inequality has two terms both of which are of the same form. For simplicity, we bound only the first one. The bound for the second one follows from the exact same steps.

The relation between sub-Gaussian and sub-exponential norms [Ver10] allows us to write

$$\| \|x\|_2 \|_{\psi_2}^2 \leq \| \|x\|_2^2 \|_{\psi_1} \leq \sum_{i=1}^p \|x_i^2\|_{\psi_1}, \tag{43}$$

$$\leq 2 \sum_{i=1}^p \|x_i\|_{\psi_2}^2 \leq 2\kappa^2 p,$$

where the second step follows from the triangle inequality. Hence, we conclude that $\|x\|_2 - \mathbb{E}\left[\|x\|_2\right]$ is a centered sub-Gaussian random variable with norm upper bounded by $3\kappa\sqrt{p}$.

For $\epsilon < 4/3$, we notice that the random variable $u_\beta(x) = g(\langle x, \beta \rangle) + \epsilon\|x\|_2/4\mu$ is also sub-Gaussian with norm

$$\|u_\beta(x)\|_{\psi_2} \leq b + \frac{\epsilon}{4\tilde{\mu}}3\kappa$$

$$\leq b + \kappa/\tilde{\mu},$$

and consequently, the centered random variable $u_\beta(x) - \mathbb{E}\left[u_\beta(x)\right]$ has the sub-Gaussian norm upper bounded by $2b + 2\kappa/\tilde{\mu}$.

Then, by the Hoeffding-type inequality for the sub-Gaussian random variables, we obtain

$$\mathbb{P}\left(\frac{1}{n}\sum_{i=1}^n u_\beta(x_i) - \mathbb{E}\left[u_\beta(x)\right] > \epsilon/2\right) \leq \exp\left\{-cn\frac{\epsilon^2}{(b+\kappa/\tilde{\mu})^2}\right\}$$

for an absolute constant $c > 0$.

By the same argument above, one can obtain the same result for the function $l_\beta(x)$. Using Hoeffding bounds in (42) along with the union bound over the net, we immediately obtain

$$\mathbb{P}\left(\sup_{\beta \in B(\tilde{\beta})}\left|\frac{1}{n}\sum_{i=1}^n g(\langle x_i, \beta \rangle) - \mathbb{E}\left[g(\langle x, \beta \rangle)\right]\right| > \epsilon\right) \leq 2\left|\mathcal{N}_{\Delta_*}\right|\exp\left\{-cn\frac{\epsilon^2}{(b+\kappa/\tilde{\mu})^2}\right\}$$

for some absolute constant $c$.

Using a standard covering argument over the net $\mathcal{N}_{\Delta_*}$ as given in Lemma 4, we have

$$|\mathcal{N}_{\Delta_*}| \leq \left(\frac{\delta\sqrt{p}}{\Delta_*}\right)^p = \left(\frac{4\delta k\tilde{\mu}p}{\epsilon}\right)^p.$$

Combining this with the previous bound, and choosing

$$\epsilon^2 = \frac{p}{n}\frac{(b+\kappa/\tilde{\mu})^2}{2c}\log\left(\frac{32c\delta^2 k^2\tilde{\mu}^2 pn}{(b+\kappa/\tilde{\mu})^2}\right)$$

we get

$$2\left(\frac{4\delta k\tilde{\mu}p}{\epsilon}\right)^p\exp\left\{-cn\frac{\epsilon^2}{(b+\kappa/\tilde{\mu})^2}\right\}$$

$$= 2\exp\left\{-\frac{p}{2}\log\log\left(\frac{32c\delta^2 k^2\tilde{\mu}^2 pn}{(b+\kappa/\tilde{\mu})^2}\right)\right\}$$

$$\leq 2\exp\left\{-p\right\},$$

whenever $np > 51\max\{\chi, \chi^{-1}\}$ for $\chi = (b+\kappa/\tilde{\mu})^2/(c\delta^2 k^2\tilde{\mu}^2)$.

$\square$

**Lemma 4** ([EM15]). *Let $B \subset \mathbb{R}^p$ be the ball of radius $\delta$ centered around some $\beta \in \mathbb{R}^p$ and $\mathcal{N}_\epsilon$ be an $\epsilon$-net over $B$. Then,*

$$|\mathcal{N}_\epsilon| \leq \left(\frac{\delta\sqrt{p}}{\epsilon}\right)^p.$$

*Proof of Lemma 4.* The set $B$ can be contained in a $p$-dimensional cube of size $2\delta$. Consider a grid over this cube with mesh width $2\epsilon/\sqrt{p}$. Then $B$ can be covered with at most $(2\delta/(2\epsilon/\sqrt{p}))^p$ many cubes of edge length $2\epsilon/\sqrt{p}$. If ones takes the projection of the centers of such cubes onto $B$ and considers the circumscribed balls of radius $\epsilon$, we may conclude that $B$ can be covered with at most

$$\left(\frac{2\delta}{2\epsilon/\sqrt{p}}\right)^p$$

many balls of radius $\epsilon$. $\qquad\square$

**Lemma 5** (Corollary 5.50 of [Ver10])**.** *Let $w_1, w_2, ..., w_n$ be isotropic random vectors with sub-Gaussian norm upper bounded by $\kappa$. Then for every $t > 0$, with probability at least $1 - 2\exp\left\{-c_1 t^2\right\}$, the empirical covariance $\widetilde{\Sigma}$ satisfies,*

$$\left\|\widetilde{\Sigma} - \mathbf{I}\right\|_2 \leq \max\{\delta, \delta^2\} \quad where \quad \delta = c_2\sqrt{\frac{p}{n}} + \frac{t}{\sqrt{n}}$$

*where $c_1, c_2$ are constants depending only on $\kappa$.*

**Remark 1.** *For $t = \sqrt{p/c_1}$, we get with probability at least $1 - 2\exp\left\{-p\right\}$,*

$$\left\|\widetilde{\Sigma} - \mathbf{I}\right\|_2 \leq C\sqrt{\frac{p}{n}}$$

*where*

$$C = \left\{c_2 + \frac{1}{\sqrt{c_1}}\right\},$$

*and $n > C^2 p$. Here, $C$ only depends on $\kappa$.*

**Lemma 6** (Corollary 5.52 of [Ver10])**.** *Let $x_1, x_2, ..., x_n$ be random vectors with mean 0 and covariance $\Sigma$ supported on a centered Euclidean ball of radius $\sqrt{R}$, i.e., $\|x_i\|_2 \leq \sqrt{R}$. For $\epsilon \in (0, 1)$ and $c > 0$ an absolute constant, with probability at least $1 - 1/p^2$, the empirical covariance matrix satisfies*

$$\left\|\widehat{\Sigma} - \Sigma\right\|_2 \leq \epsilon \left\|\Sigma\right\|_2,$$

*for $n > cR\log(p)/(\epsilon^2 \left\|\Sigma\right\|_2)$.*