[Reviews · NeurIPS 2016]

Reviewer 1

Summary

The authors investigate estimating GLMs when n is much larger than p. They exploit the fact that the GLM estimator is sometimes proportional to the OLS estimator and show this holds more generally for random design problems. They propose a two step estimator: firstly a fast OLS estimator then using a fast root finder to estimate the constant of proportionality. Finally on benchmark datasets they show their method reaches a desired test error much quicker than a large variety of standard first and second order optimisation algorithms.

Qualitative Assessment

Overall I think the contribution is nice. The paper is well written and presented. The main idea is a good one and appears novel. That said, I have the following concerns: The experiments seem extensive but there is a glaring omission. A stochastic optimisation method should have been used for comparisons. For example I have a strong feeling that SVRG [Johnson&Zhang 2013] would have performed better than most of the other competing methods. The theorems and in particular prop 2 only hold for sub-gaussian random variables. The results are nice but the naive subsampling estimator will certainly perform badly when the data has heavier tails. This is why Dhillon et al and many others have proposed random-projection based estimators. Although using these will immediately add an extra factor of O(np log p) to the computation time. It would have been nice to see experiments with heavier tailed data to demonstrate the robustness of the algorithm. Therefore, it is not clear exactly how useful the algorithm is as it is presented. A practitioner would have to exercise care when dealing with potentially heavy-tailed data but might anyway get the same speedups (without having to be as vigilant) by using a stochastic variance reduced optimisation method.

Confidence in this Review

2-Confident (read it all; understood it all reasonably well)


Reviewer 2

Summary

The authors argue that in the n >> p settings GLMs parameters can be approximately estimated by a scaled OLS estimator. The authors present an algorithm that exploits this phenomenon and demonstrate experimentally that it can estimate GLM parameters faster than a number of alternative algorithms and in a number of GLMs/datasets. Additionally, the authors provide theoretical arguments that this phenomenon extends beyond simple Gaussian designs.

Qualitative Assessment

Finding efficient surrogates to large-scale optimization problems is an important problem and the authors point out an elegant method for estimating GLMs when n >> p. As acknowledged by the authors the fundamental ideas relating to the Gaussian case are not new, but the extension to non-Gaussian designs is valuable and interesting. Overall, I think the paper is well-written and clearly organized. Main feedback: - The results in Figure 1 should be extended with a plot of time vs accuracy for the two methods, as done in Figure 2. It is clear that solving the SLS optimization problem should be faster than the corresponding GLM (e.g., Figure 1a), and that at the same time we encur a cost in accuracy (e.g., Figure 1b). - The inequality of Theorem 1 does not appear to be tight. I.e., it doesn't recover the result of Proposition 1 in the special case when the covariates are Gaussian. It would be helpful if the authors could comment on this gap. - It isn't entirely clear when the authors use |S| = n (i.e., in Figure 1, I presume) and when |S| \leq n. E.g., what was the choice of |S| in the experiments of Fig 2, Table 1? Minor: - Line 159: Hayley -> Halley - Proposition 2 and Theorem 2: The authors didn't define \lambda_min in the main paper.

Confidence in this Review

2-Confident (read it all; understood it all reasonably well)


Reviewer 3

Summary

This paper exploits a simple proportionality relationship between ordinary least squares (OLE) problems and Generalized Linear models (GLMs). By solving OLE first, followed by estimation of the constant of proportionality, the paper comes up with faster GLM solvers. The theoretical part of the paper proves that under reasonable assumptions, such a procedure will estimate the GLM parameters accurately in the n much bigger than p setting.

Qualitative Assessment

Interesting paper - the reduction of GLM to OLE is striking. For what class of regularizers (section 3.1) does the proportionality relationship continue to hold? What classes of link functions are particularly favorable to this reduction? Section 5: Speedup results are unsurprising. It is clear that an OLE approach + single parameter estimation of the proportionality constant would be faster than iterative optimization of the GLM objective function. More interesting to the reviewer is the question of how far this idea can be pushed beyond the exact equivalence of Proposition 1. In Figure 2, one would expect the SLS solution to be slightly worse than the GLM solution. But this is consistently not the case, which is quite striking. Does it imply that on these datasets and regime for samplesize "n", the error (Eqn 10, Eqn 14) would be very smallish?

Confidence in this Review

2-Confident (read it all; understood it all reasonably well)


Reviewer 4

Summary

The paper is based on the fact that, when the predictors are normally distributed, the regression coefficients of a GLM (with canonical link) are proportional to the corresponding GLM coefficients. The authors propose to exploit this fact to reduce the computational burden of fitting GLMs. In particular, they estimate the coefficients by OLS and then they find the proportionality constant using an univariate root-finding method. They also show that the results approximately apply outside the Gaussian context, and provide bounds on the discrepancy between GLM and (scaled) OLS coefficients.

Qualitative Assessment

The main idea of the paper is an interesting and, at least to me, novel one. Sections 1 to 3 are extremely well written, while the second half of the paper seems a bit more hurried. It would be worthwhile pointing out early in the paper that the proposed approach applies only when the canonical link is used (currently stated in line 92). However, the authors say: "Motivated by the results in the previous section, we design a computationally efficient algorithm for any GLM task that is as simple as solving the least squares problem; it is described in Algorithm 1." The "any" suggests that the canonical link is not required. Please clarify. In the right plot of Figure 1 it might be more useful plotting the relative accuracy of the proposed method, that is MSE(SLS)/MSE(MLE). Also, are you using sub-sampling or the full sample for SLS here? In Figure 1 STS does strictly worse than MLE in terms of accuracy. Given that here the design is Gaussian and given Proposition 1, it is not entirely clear to me why this should happen. It is because estimating the constant c introduces extra variance? I am a bit puzzled by the decision of using sub-sampling here. The STS method should be applicable whether or not subsampling is used. Hence using subsampling makes the results in Section 5 harder to interpret. That is, in terms of speed-up and accuracy what is the contributions of STS and what is that of subsampling? L204 The authors say: "On the other hand, each of the MLE algorithms requires some initial value for , but no such initialization is needed to find the OLS estimator in Algorithm 1. This raises the question of how the MLE algorithms should be initialized, in order to compare them fairly with the proposed method." It would have been useful to include also Iteratively Reweighted Least Squares (IRLS) in this comparison, given its popularity. Also, Wood (2006) page 66 explains that a default initialization is available for this method. L215 The authors say: " For each dataset, the minimum achievable test error is set to be the maximum of the final test errors, where the maximum is taken over all of the estimation methods." Using this approach, if one of the methods fails, the test error will be set to a fairly high number. Have all methods converged on all datasets? In Section 5 the authors use two simulated datasets. Are the results in Figure 2 and Table 1 the output of a single run and an average of several runs? At least a handful of run are needed in order to provide some confidence in the results. Figure 2 is quite crammed. Maybe some space could be saved by eliminating redundant legends and axes labels. The lines in the legend are extremely small. Maybe it is also worth pointing out in the caption that STS does not start from zero on the x axis because of the fixed cost paid by OLS. Lines 253-257 are quite confusing. What is the take home message of Corollary 1? I don't immediately see what the lasso results implies for the unpenalized GLM regression. Corollary 1 makes the assumption beta > eta / s, what is the meaning of this assumption and is it realistic? Similarly the authors set lambda = eta / (c*s), why this choice? Lambda_min seems to be undefined in Proposition 2. It this defined in the Appendix, but something about it must be said also in the main text. MINOR POINTS: L46 I am not sure about this sentence: "For logistic regression with Gaussian design (which is equivalent to Fisher’s discriminant analysis)". Friedman et al. (2008) on page 127 say that they are quite different, even with Gaussian design. In particular, discriminant analysis is more efficient in this case, because it exploits the normality of the predictors. L161 "that we can attain up to a cubic rate of convergence (by, for instance, using Hayley’s method)" this is a repetition. L296 "Another interesting line of research is to find similar proportionality relations between the parameters in other large-scale optimization problems such as support vector machines. Then the problem complexity can be significantly reduced as in the case of GLMs." The latter sentences should be conditional on the former, but at the moment it is an assertion. REFERENCES: - Wood, Simon. Generalized additive models: an introduction with R. CRC press, 2006. - Friedman, Jerome, Trevor Hastie, and Robert Tibshirani. The elements of statistical learning. 2nd Edition. Springer, Berlin: Springer series in statistics, 2008.

Confidence in this Review

2-Confident (read it all; understood it all reasonably well)


Reviewer 5

Summary

This paper studies the parameter estimation problem of generalized linear models when the number of observations n is much larger than the number of predictors p. An algorithm is proposed to estimate parameters based on an observation that GLM parameters are approximately proportional to the solution of OLS.

Qualitative Assessment

This paper studies the parameter estimation problem of generalized linear models when the number of observations n is much larger than the number of predictors p. An algorithm is proposed to estimate parameters based on an observation that GLM parameters are approximately proportional to the solution of OLS. The paper is well-structured and well-written. The derivations look correct to me. The authors also demonstrated through experiments that the proposed algorithm outperforms the standard methods by a large margin. It is a solid paper and therefore I recommend the acceptance.

Confidence in this Review

2-Confident (read it all; understood it all reasonably well)


Reviewer 6

Summary

This paper proposes an efficient algorithm to solve GLMs in large-scale problems under certain random designs. It can attain cubic convergence after an approximate OLS estimator is found, with only O(n) per-iteration computation. Theoretical analysis gives convergence rate of the estimation error and numerical experiments demonstrate the results.

Qualitative Assessment

An old idea of relating GLM coefficient to OLS coefficient is applied to modern large-scale settings, assuming the OLS estimator can be obtained accurately with far less computation. The method is novel. It also generalizes the theoretical results to non-Gaussian design based zero-bias transformation. Typo: cubic convergence - Do you mean Halley’s method? I have one question regarding the subsampling step in obtaining OLS estimator. One alternative could be do this step at the very beginning, and then either compute the MLE directly, or follow the similar procedure proposed here. How would that compare to the current results? In general, this paper is very well written and has many interesting results and implications.

Confidence in this Review

2-Confident (read it all; understood it all reasonably well)